# One-atom-thick hexagonal boron nitride co-catalyst for enhanced oxygen evolution reactions

Yizhen Lu[1], Bixuan Li[2,3], Na Xu[1], Zhihua Zhou[1], Yu Xiao[1], Yu Jiang[1], Teng Li[1], Sheng Hu ●[1,4,5], Yongji Gong ●[2,6] ✉ & Yang Cao ●[1,4,5] ✉

Developing efficient (co-)catalysts with optimized interfacial mass and charge transport properties is essential for enhanced oxygen evolution reaction (OER) via electrochemical water splitting. Here we report one-atom-thick hexagonal boron nitride (hBN) as an attractive co-catalyst with enhanced OER efficiency. Various electrocatalytic electrodes are encapsulated with centimeter-sized hBN films which are dense and impermeable so that only the hBN surfaces are directly exposed to reactive species. For example, hBN covered Ni-Fe (oxy) hydroxide anodes show an ultralow Tafel slope of ~30 mV dec$^{-1}$ with improved reaction current by about 10 times, reaching ~2000 mA cm$^{-2}$ (at an over-potential of ~490 mV) for over 150 h. The mass activity of hBN co-catalyst is found exceeding that of commercialized catalysts by up to five orders of magnitude. Using isotope experiments and simulations, we attribute the results to the adsorption of oxygen-containing intermediates at the insulating co-catalyst, where localized electrons facilitate the deprotonation processes at electrodes. Little impedance to electron transfer is observed from hBN film encapsulation due to its ultimate thickness. Therefore, our work also offers insights into mechanisms of interfacial reactions at the very first atomic layer of electrodes.

Water electrolysis is a sustainable and clean strategy to convert electrical energy into chemical fuels, but its efficiency is considered to be limited by the oxygen evolution reaction (OER) at anodes with respect to the cathode reaction of hydrogen evolution[1,2]. This is because of the 4-electron transfer process at the interface between electrodes and electrolytes, where a multistep proton-electron exchange is involved in the transfer of each electrons. That usually leads to the sluggish OER kinetics[3,4]. In addition, multiple oxygen-containing intermediates (e.g., OH*, O*, and OOH*) are also found to be incorporated in the interfacial adsorption/desorption procedures[5,6], which further complex the mechanism understanding of OER reactions. To improve OER

performance, extensive efforts have been devoted to developing electrocatalysts that contain active sites where electrons are localized to facilitate/optimize the adsorption of reactive species[7–9]. Such electrocatalysts usually have defective sites (including edge sites, atomic steps, and doping sites), or consist of heteroatoms to provide chemical affinity to species[7,10,11]. However, the defective sites usually contain active dangling bonds that may influence the stability of catalysts[12,13]. Furthermore, localized electrons are in general associated with impeded electron transport through catalysts' bulk to electrochemical circuits. This tradeoff between conductivity and activity motivates the search for novel catalytic materials with optimal tradeoff characteristics[14]. Another

[1]State Key Laboratory of Physical Chemistry of Solid Surfaces, Collaborative Innovation Center of Chemistry for Energy Materials (iChEM), College of Chemistry and Chemical Engineering, Xiamen University, Xiamen 361005, China. [2]School of Materials Science and Engineering, Beihang University, Beijing 100191, China. [3]School of Physics, Beihang University, Beijing 100191, China. [4]Innovation Laboratory for Sciences and Technologies of Energy Materials of Fujian Province (IKKEM), Xiamen 361005, China. [5]Pen-Tung Sah Institute of Micro-Nano Science and Technology, Xiamen University, Xiamen 361005, China. [6]Tianmushan Laboratory, Hangzhou 310023, China. ✉e-mail: yongjigong@buaa.edu.cn; yangcao@xmu.edu.cn

possible route is the assembly of co-catalysts on catalytic electrodes to form heterogeneous structures and improve the mass and energy transfer process at interfaces[15,16]. A recent progress is to use molecular-scale ligands as co-catalysts to further reduce the distance of electron and proton transport from electrolyte to electrodes[16]. Nevertheless, the stability of co-catalysts is usually influenced by their contacts with electrodes, and their coverage is expected to reduce the effective activation area of electrodes[15,17].

For this task, atomically-thin two-dimensional (2D) materials are possible candidates. When being integrated between reactant and electrochemical electrodes, 2D materials shorten the electron transport path due to their atomic-scale thickness and thus reduce the impedance of electron transfer. In addition, 2D materials are capable of forming heterostructures with other zero-dimensional (0D), one-dimensional (1D), 2D, and three-dimensional (3D) catalytic electrodes via van der Waals interactions[18,19]. Such interactions are free of dangling bonds and allow efficient tuning of interfacial properties. That greatly extends the possibility of creating functional interfaces on demands[20,21]. Nonetheless, further improving OER performances of 2D materials and their heterostructures requires a better understanding of the underlying mechanisms. That goal can only be achieved in 2D catalytic systems that are controllably fabricated. Previous experimental studies were mostly based on 2D nanosheets with unevenly distributed thicknesses and lateral sizes, which complexes the quantitative analysis of mass and charge transfer at interfaces[22–24]. Furthermore, the basal plane of 2D materials is conventionally considered electrochemically inert[25,26]. Until recently, it has been reported that such a plane can be electrochemically activated via heterogeneous assembly. Insight exploration in this area also requires 2D systems to have well-defined adsorption sites and uniform thickness, which properties still remain challenging to realize.

In this work, we present a class of OER anodes where electrocatalytic electrodes are covered by centimeter-sized hexagonal boron nitride (hBN) films. The unique feature of hBN is its chemical affinity to oxygen-containing species, due to the strongly polarized B-N bond with valence electrons localized around N atoms[27,28]. It has been reported that such affinity optimizes the adsorption of various oxygen-containing species and thus enhances catalytic performances[29,30].

From that perspective, each B-N bond can be considered as an adsorption site, which can be translated to a site density ~$10^{15}$ cm$^{-2}$ with all sites exposed on the surface. Despite being an insulator in-plane, our hBN crystals may not necessarily impede interfacial charge transfer in the direction perpendicular to the plane. This is because of their mono-atomic-scale thickness that allows for fast electron tunneling as an additional charge transport mechanism[31]. All these characteristics provide hBN as a potential co-catalyst to modify interfacial properties and thus OER activity of electrodes.

## Results

### Synthesis and structural characterization

Figure 1 explains the structure of our OER anodes. Fabrication procedures are detailed in Methods. Briefly, one-atom-thick hBN films were synthesized on Cu substrates using chemical vapor deposition (CVD) technique through a surface-mediated growth mechanism[32] (Fig. 1a, b). The resulting film is polycrystal, as being indicated by the electron microscopy (Fig. 1c). More characterization about the hBN layer's thickness, band gap, and elemental analysis can be found in Supplementary Fig. 1. Ion and gas permeation experiments further suggest that the monolayer hBN is dense, impermeable to OH$^-$ ions (Supplementary Fig. 2) and even the smallest of helium gas (kinetic diameter ~2.6 Å. Supplementary Fig. 3). Within our measurement accuracy limit, we estimate a membrane porosity <$10^{-6}$, or 1 nm$^2$ defective area per micron meter square. Subsequently, the hBN film is transferred to cover the target OER electrode which consists of a catalytic layer electrochemically deposited on Au contacts, as shown in Fig. 1d–g. Detailed transfer procedures can be found in Methods and Supplementary Fig. 4. To start with, we choose Ni-Fe (oxy)hydroxides (NiFeO$_x$H$_y$) as the catalytic layer, which is the most active catalysts for OER in basic environments[33]. Detailed preparation methods of the NiFeO$_x$H$_y$ layer can be found in Methods and the related characterization using X-ray photoelectron spectroscopy (XPS) in Supplementary Fig. 5. Note that the NiFeO$_x$H$_y$ layer we prepared is amorphous (Supplementary section "Characterization of NiFeO$_x$H$_y$ catalytic layer"), which is expected to have improved catalytic performance compared with crystalline NiFeO$_x$H$_y$ due to the more active sites[34]. The hBN-NiFeO$_x$H$_y$ heterostructure is bonded via van der Waals forces.

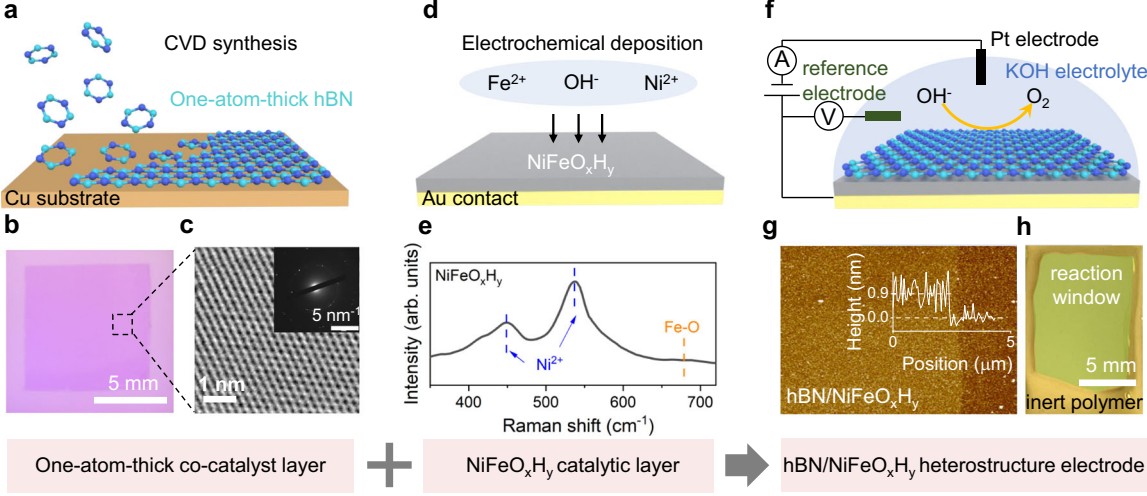

**Fig. 1 | One-atom-thick co-catalyst covered OER electrodes. a–c** Fabrication of monolayer hexagonal boron nitride films. **a** Schematic of the chemical vapor deposition (CVD) method. **b** Optical image of a centimeter-sized film on a Si/SiO$_2$ substrate. **c** High Angle Angular Dark Field-Scanning Transmission Electron Microscopy (HAADF-STEM) image of a monolayer hBN crystal. Inset shows the Electron diffraction patterns of our hBN film, indicating its polycrystal nature. **d–e** Fabrication of the NiFeO$_x$H$_y$ catalytic layer on Au contacts. **d** Schematic of the

electrochemical deposition method. **e** Raman spectrum of the resulting layer. Two characteristic peaks of Ni$^{2+}$ at 450 cm$^{-1}$ and 540 cm$^{-1}$ are observed[55]. The broad Raman peak at 680 cm$^{-1}$ can be attributed to Fe−O bonds in the NiFeO$_x$H$_y$[56]. **f–h** OER electrodes fabricated by covering the NiFeO$_x$H$_y$ catalytic layer using one-atom-thick hBN. **f** Schematic of our heterostructure electrodes and measurement set up. **g** Height profile of the hBN layer on NiFeO$_x$H$_y$. **h** Optical image of a final device.

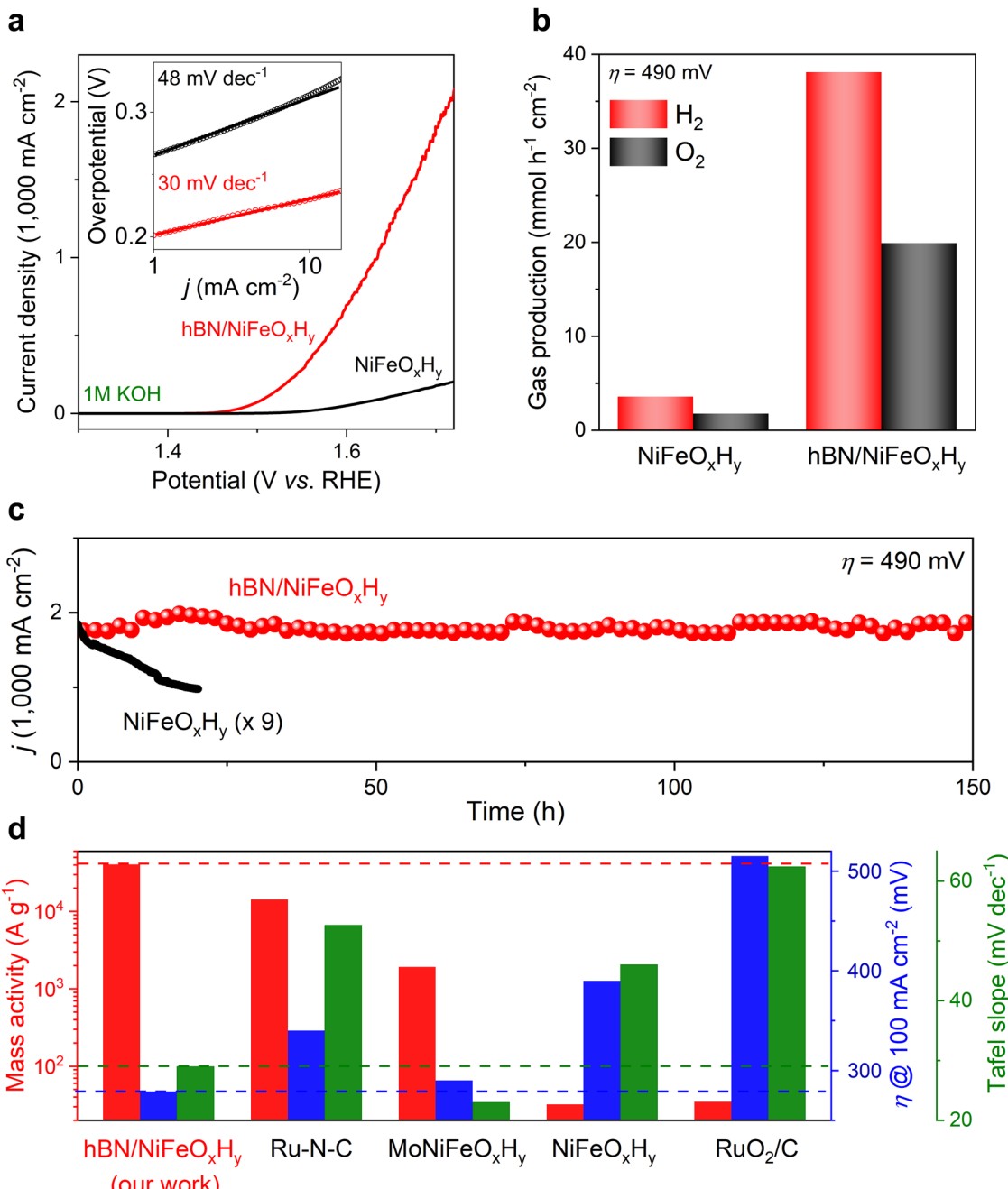

**Fig. 2 | OER performance of our heterostructure electrodes. a** Linear sweep voltammetry (LSV) of a representative hBN/NiFeO$_x$H$_y$ electrode. For comparison, the result of a bare NiFeO$_x$H$_y$ electrode is also presented. Inset shows their Tafel slopes. **b** Gas product analysis at $\eta$ = 490 mV for the hBN/NiFeO$_x$H$_y$ electrode and the bare NiFeO$_x$H$_y$ electrode. **c** Stability of our hBN/NiFeO$_x$H$_y$ electrodes. For comparison, results from the bare NiFeO$_x$H$_y$ electrode are multiplied by nine times. **d** Performance comparison of our hBN co-catalyst (on NiFeO$_x$H$_y$) with other state-of-the-art catalysts reported. To find the mass activity of hBN/NiFeO$_x$H$_y$, we estimate the mass of hBN using its known density and area, and estimate the mass of NiFeO$_x$H$_y$ by the quantity of electric charge deposited. The mass loading and resistance of hBN/NiFeO$_x$H$_y$ is 4.6 µg and 0.4 Ω cm$^{-2}$ (1.53 V versus RHE), respectively. The pH value of KOH solution is 13.65. Detailed performance comparison can be found in Supplementary Table 1.

This is also supported by our XPS measurements where no detectable chemical shift of Ni, Fe, B, and N signals are observed on hBN and NiFeO$_x$H$_y$ before and after heterostructure assembly (Supplementary Fig. 6).

**OER performance of hBN/NiFeO$_x$H$_y$ heterostructure electrodes**

We evaluate the OER performance of our electrodes in KOH solutions. The electrode is partially encapsulated with electrochemically inert polymers as such only the hBN-covered areas remain exposed and partake in reactions (Fig. 1h). A bias potential $V_{RHE}$ (potential with respect to the reversible hydrogen electrode, RHE) is applied between the Au contact and a Pt electrode with a surface area large enough not to limit reaction rates. Figure 2a shows a typical current-voltage characteristic for the OER. The recorded current density $j$ reaches 10 mA cm$^{-2}$ at an overpotential $\eta$ of 230 mV (where $\eta = V_{RHE}-1.23$ V, the theoretical equilibrium potential for water electrolysis), and increases quickly with $V_{RHE}$ at a slope of close to 30 mV dec$^{-1}$ (which is also known as the Tafel slope). As a result, the $j$ reaches up to 2000 mA cm$^{-2}$ at $\eta \approx 490$ mV. To the best of our knowledge, such a $j$ value is among the highest of those being reported previously. Importantly, our

electrode is highly stable even at high $j$. The current varies less than 10% at $j = 2000$ mA cm$^{-2}$ within 150 h (Fig. 2c). Note that no morphology or structural change of hBN has been observed before and after OER (see electron microscope and X-ray spectrum characterization in Supplementary Fig. 7), confirming the stability of hBN encapsulation. For comparison, we also measured the OER performance of bare NiFeO$_x$H$_y$. The current density is about 10 times lower than that from electrodes covered with hBN films, and decays more than 30% within 10 h. Note that such performance of bare NiFeO$_x$H$_y$ is comparable to that from literature[35]. Many realistic applications require OER current densities >1000 mA cm$^{-2}$ [2,36], and our electrode easily meets that standard at a relatively low $\eta \approx 400$ mV. The effective electrochemical active surface area (ECSA) of the hBN/NiFeO$_x$H$_y$ is measured to be ~3 cm$^2$ (Supplementary Fig. 8). That leads to a current density normalized by ECSA an order of magnitude higher than other NiFeO$_x$H$_y$ catalysts in literature[37]. Detailed comparison of the OER performance can be found in Fig. 2d and Supplementary Table 1. Gas chromatography results (Fig. 2b) further show that the reaction products are O$_2$ and H$_2$ (molar ratio 1:2), with a Faraday efficiency >99% (Supplementary Table 2 for faraday efficiency estimation). The production of gases on hBN/NiFeO$_x$H$_y$ electrodes is also about 10 times faster than that of electrodes without hBN coverage, which result is consistent with Fig. 2a. Due to the high density of surface adsorption sites and the atomically thin nature of the hBN film, its mass activity is found to be extremely high, reaching ~10$^6$ A g$^{-1}$ at $\eta = 300$ mV (Supplementary Table 1). The hBN/NiFeO$_x$H$_y$ mass activity is estimated ~10$^4$ A g$^{-1}$ (Fig. 2d). This is three orders of magnitude higher than that of commercial Ru-based catalyst, and exceeds that from other state-of-the-art catalysts reported; a performance comparison of mass activities is shown in Fig. 2d and Supplementary Table 1.

We attribute the enhanced stability to the impermeable hBN layer encapsulation[38], which blocks the access of reactive species to the NiFeO$_x$H$_y$ surface and prevents any local etching process there[39]. The hBN encapsulation is also found to prevent the lattice oxygen in the NiFeO$_x$H$_y$ from participating in OER[40]. To that end, we prepare hBN/NiFeO$_x$H$_y$ electrodes where the NiFeO$_x$H$_y$ is labeled with $^{18}$O (Supplementary Fig. 9). No $^{18}$O element is detected in the O$_2$ products within our detection limit. In a parallel experiment, adding TMAOH (Tetramethylammonium hydroxide) that is expected to competitively occupy lattice oxygen sites at NiFeO$_x$H$_y$ surfaces with respect to that of OH species[41] has no influence on the observed OER performance (Supplementary Fig. 10). All these experiments are consistent with the explanation of electrode stability, and suggest that OER-related species react on hBN surfaces without any direct contact with NiFeO$_x$H$_y$ underneath. To further prove the importance of hBN encapsulation to the OER performance, we use monolayer graphene instead of hBN to cover the NiFeO$_x$H$_y$ surface and find little changes of OER efficiency (within twice variation in current density, Supplementary Fig. 11). This result indicates that the high $j$ is closely related to the surface properties of the very first atomic layer covered at electrodes, and the layer's electrical conductivity does not play any notable role.

## Mechanisms of hBN film catalyzed OER reactions

To reveal the OER mechanism behind, we first study the influence of defects on hBN layers to the OER performance. To that end, we use mechanically exfoliated hBN monolayer crystals, which are known to have negligible number of defects[42], to cover the NiFeO$_x$H$_y$ electrodes (Supplementary Fig. 12). We find an enhanced OER current of ~2 A cm$^{-2}$ at $V_{RHE} = 1.72$ V (Supplementary Fig. 13), which performance is similar to that from the CVD hBN/NiFeO$_x$H$_y$ samples. This enhancement indicates that defects in hBN layers are not likely to be the reactive sites while the basal plane is electrocatalytic. Next, we explore the rate-determining step by measuring the OER activity as a function of solution pH (from pH ~11–14). Figure 3a shows that the log $j$ (at $V_{RHE} = 1.72$ V) increases proportionally with pH, from which the proton reaction orders on RHE

scale ($\rho^{RHE} = \partial \log j / \partial pH$) is found close to 1. The strong pH dependence indicates that the concentration of either hydroxide ions or protons are critical to the OER activity. To find out the answer, we evaluate the performance of our hBN/NiFeO$_x$H$_y$ electrode in either KOH (dissolved in H$_2$O) or KOD (dissolved in D$_2$O) solutions. A noticeable isotope effect is observed as shown in Fig. 3b. For example, the current density (at $V_{RHE} = 1.72$ V) obtained in KOH is about twice higher than that in KOD. Such a large difference cannot be attributed to hydroxide ions because of the very similar atomic mass and chemical properties between OH$^-$ and OD$^-$. Thus, we conclude that proton transfer is involved in the rate-determining step of OER reactions.

Based on the above understandings, we perform density functional theory (DFT) calculations to identify OER mechanisms at hBN/NiFeO$_x$H$_y$ electrodes. Details of our calculations can be found in Supplementary (Supplementary section 'DFT analysis'; Supplementary Figs. 14 and 15, Supplementary Table 3 and Supplementary Table 4). The distance between the hBN and NiFeO$_x$H$_y$ layer is calculated to be 0.40 nm, in consistent with the typical van der Waals distances of 0.33–0.84 nm[18,43]. We find a noticeable downward shift of the density of states for the anti-bonding orbitals of hBN after being incorporated with NiFeO$_x$H$_y$. As a consequence, charge transfer (-0.15 $e^-$ per 1.33 nm$^2$, which is the area used in our model that contains 25 units of hBN and 16 units of Ni$_{0.75}$Fe$_{0.25}$O$_x$H$_y$) from hBN to NiFeO$_x$H$_y$ occurs at the heterostructure electrode, which builds up an interlayer electric field (Fig. 3c top inset and Supplementary Table 3). That causes enhanced adsorption of OH* at B sites with an adsorption energy close to 1.8 eV with respect to that of -0.9 eV on bare hBN (Supplementary Table 4). Such adsorption also leads to a greater electronic displacement of O atoms toward B atoms[44]. Therefore, interactions between O and H atoms in OH* species are weakened, which behavior considerably reduces the deprotonation barrier from OH* to O*. Note that the result of OH* deprotonation as the rate-determining step is also consistent with Fig. 3a, b.

Now we turn to electron transport processes in our electrodes. We emphasis that the electrical resistance of monolayer hBN does not restrict the OER activity in our case. Otherwise, the $\rho^{RHE}$ in Fig. 3a should deviate from 1 because the supply of electrons cannot support the fast reactant conversion at high pH, in contrast to experimental observations. The Electrochemical Impedance Spectroscopy results (Supplementary Fig. 8) and conductivity characterization of hBN films (Supplementary Fig. 16) further supports the low interfacial resistance introduced by hBN encapsulation. Unfortunately, the electric-resistance-independent reactions also restrict us from finding out the electron transport mechanism through monolayer hBN films. One possibility is fast electron tunneling through atomically thin hBN layers[31,45,46]. This is evidenced by the decreased $j$ found on thicker hBN encapsulated devices (Supplementary Fig. 17) because of the increased electron tunneling barrier that scales exponentially with thickness. On the other hand, in situ Raman spectroscopy results (Fig. 4) show electron transfer in the NiFeO$_x$H$_y$ layer via the valence change of Ni elements. The conversion of Ni$^{2+}$ to Ni$^{3+\delta}$ for hBN/NiFeO$_x$H$_y$ electrodes starts at about $V_{RHE} = 1.08$ V versus RHE, and is significantly lower than that of 1.38 V using bare NiFeO$_x$H$_y$ electrodes (Fig. 4). This result is also in agreement with the lower overpotential found at hBN/NiFeO$_x$H$_y$ electrodes (Fig. 2a). To understand the roles of hBN and NiFeO$_x$H$_y$ components in the heterostructure, we measure the XPS spectrum of both the hBN/NiFeO$_x$H$_y$ and bare NiFeO$_x$H$_y$ electrodes before and immediately after OER reactions. As shown in Supplementary Fig. 18, the valence state changes of Ni and Fe elements are identical in the two cases, respectively. Therefore, we conclude that Ni and Fe elements synergistically interact and have similar functions as those in bare NiFeO$_x$H$_y$ electrodes. The latter functions are widely reported in literatures[47,48]. To explain the observed conversion behaviors in Fig. 4, we recall the interlayer electrical field originated from the charge transfer between hBN and NiFeO$_x$H$_y$ layers. That effectively promote

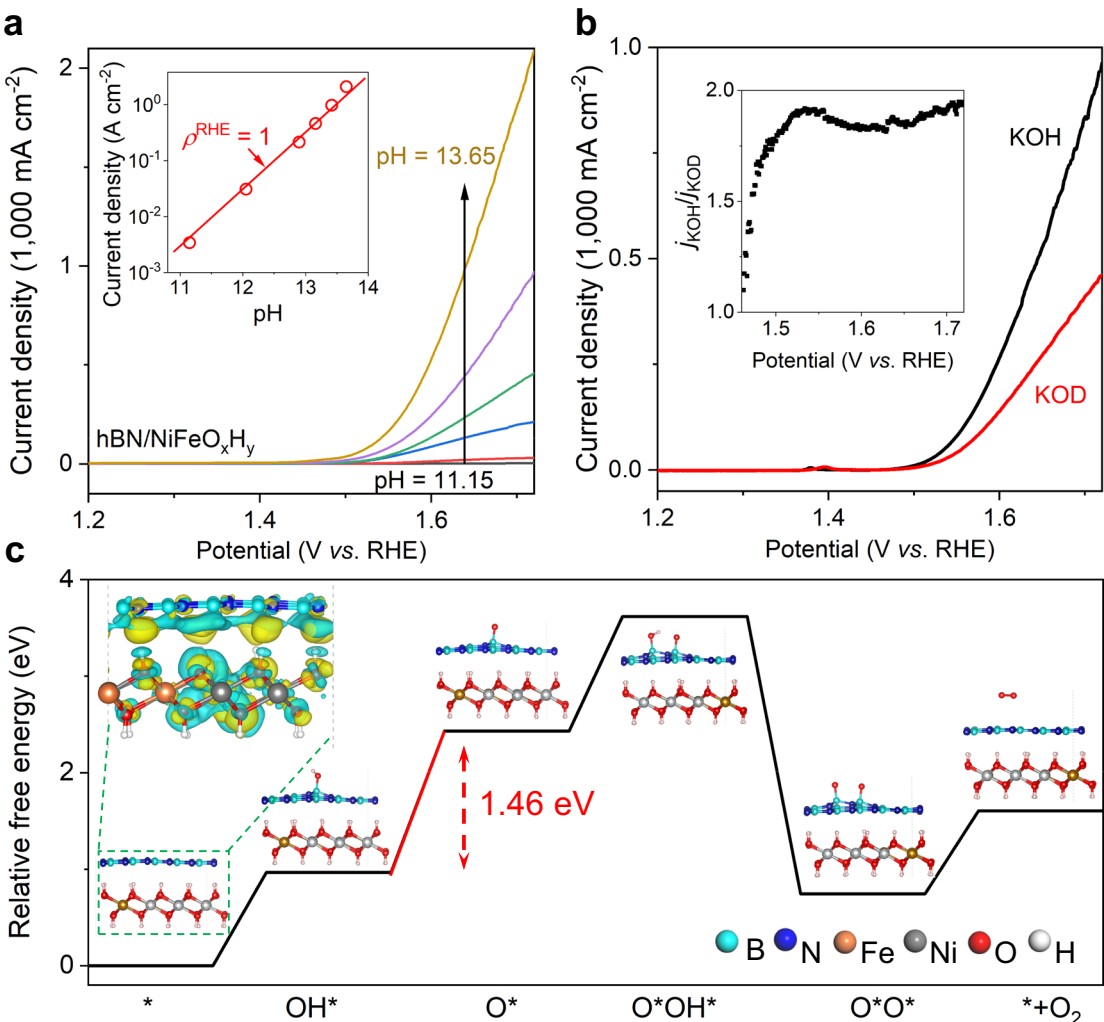

**Fig. 3 | Mechanisms of hBN film catalyzed OER reactions. a** Current density measured in KOH with pH = 11.15, 12.05, 12.90, 13.16, 13.42, and 13.65, for curves from bottom to top. These pH values correspond to KOH solution concentrations of 0.001 M, 0.01 M, 0.1 M, 0.2 M 0.5 M, and 1 M, respectively. Inset shows OER current density at 1.72 V versus RHE plotted in log scale as a function of pH. The red line shows the proton reaction order prediction using $\rho^{RHE} = \partial \log j/\partial pH = 1$. **b** LSV curves for hBN/NiFeOₓHᵧ electrodes measured in 1 M KOH (dissolved in H₂O) and KOD (dissolved in D₂O) solutions. Inset shows the kinetic isotope effect. $j_{KOH}$ and $j_{KOD}$ are the current densities measured in KOH and KOD, respectively. **c** Gibbs free energy diagrams of the hBN catalyzed OER reactions. Top left inset shows the charge transfer between hBN and NiFeOₓHᵧ layers, with yellow and cyan clouds showing electron accumulation and depletion near atoms, respectively. Other insets show the simulated reaction processes. The rate-determining step is marked by the red line.

the change of Ni's valence state at lower over potentials. Such results also suggest the importance of the heterogeneous assembly between hBN and NiFeOₓHᵧ in enhanced OER performance.

To summarize, we introduce one-atom-thick hBN film as an efficient co-catalyst for OER reactions. Despite being an insulator, hBN shows little impedance to electron transfer due to its ultimate thickness. The strong adsorption of oxygen-containing intermediates at the co-catalyst facilitates the deprotonation processes and enhances the interfacial activity at electrodes. Thus, our hBN co-catalyst provides enhanced OER currents up to 2000 mA cm⁻² and mass activity that is orders of magnitude larger than other catalysts. It is surprising that changing the very first atomic layer at electrode surface improves efficiency by more than 10 times. In Supplementary Fig. 19, we further demonstrate that hBN encapsulation is a universal strategy of improving OER performance on various electrodes. Those electrodes include metal (oxy)hydroxide, metal, and carbon, with enhanced OER currents up to ~20 times, depending on the adsorption energies of OH* species on them. Future optimization directions of using hBN co-catalysts for practical applications could be the development of large-scale hBN synthesis methods and autonomous transfer techniques

with improved reproducibility and efficiency[49,50]. Developing various synthesis methods such as wet chemistry methods may also reduce the cost of hBN films and thus promote their application[51]. Our results also indicate that the inert basal plane of 2D materials can be catalytically active if combined with other nanomaterials to form heterostructures. Together with the mechanism understanding of the role of oxygen species adsorption and electron transport path at heterostructures, this work can guide the rational design of 2D material-based non-precious metal catalysts with optimal performance.

## Methods

### Synthesis of one-atom-thick hBN

Our one-atom-thick hBN film was synthesized using CVD technique. First, the commercially available polycrystalline Cu foils were electro-chemically polished to remove surface contaminants and smooth the surface. The electropolishing solution consisted of 200 mL deionized water, 100 mL ortho-phosphoric acid, 100 mL ethanol, 20 mL iso-propanol, and 2 g urea. The Cu foil, clamped by an alligator clip, was immersed in the solution as an anode. Another large-sized Cu plate served as the cathode. A constant voltage of 4.80 V was applied for

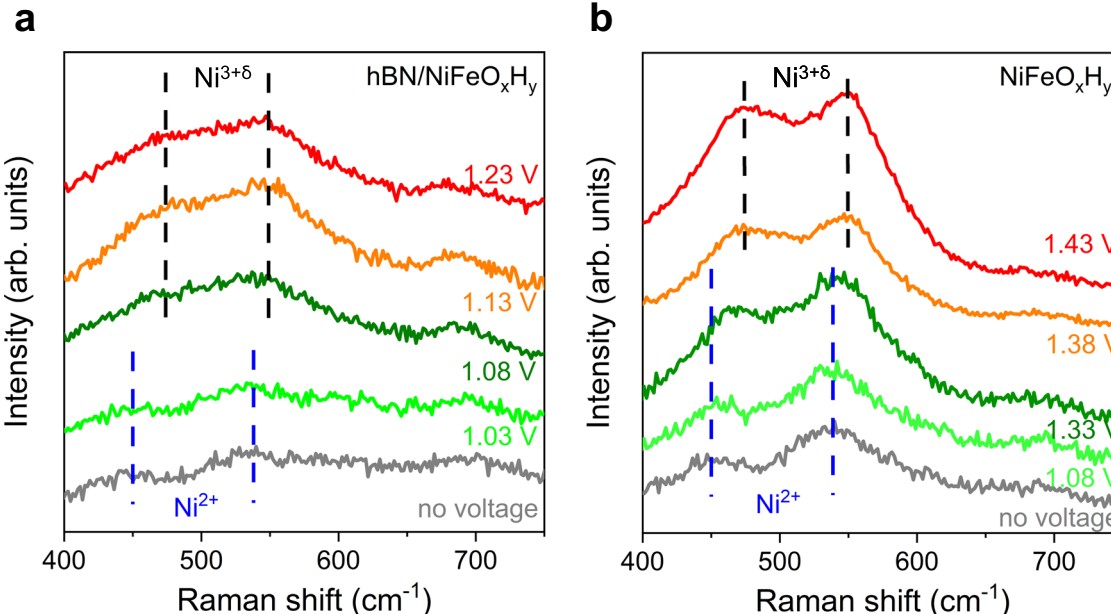

**Fig. 4 | Electron transport in hBN/NiFeOₓHᵧ heterostructures. a** and **b** are in situ Raman spectra of hBN/NiFeOₓHᵧ and NiFeOₓHᵧ electrodes at various potentials versus RHE, respectively. The blue and black dashed lines mark the characteristic peaks for $Ni^{2+}$ (at 450 cm⁻¹ and 540 cm⁻¹) and $Ni^{3+\delta}$ (at 470 cm⁻¹ and 550 cm⁻¹), respectively[55–57].

2 min. Following electropolishing, the Cu foil was rinsed with deionized water and ethanol, then dried at 60 °C in a vacuum. The electropolished Cu foil was subsequently positioned at the center of a 1-inch quartz tube, heated to 1050 °C for 40 min, and annealed for an additional 30 min at the same temperature in an Ar/H₂ atmosphere (180 sccm Ar and 20 sccm H₂). Upon annealing, the atmosphere and temperature were maintained, while 50 mg of ammonia borane was positioned at the periphery of the high-temperature zone within the furnace at 70–90 °C. The deposition process lasted between 10 and 40 min, yielding our continuous one-atom-thick hBN films. After the reaction, ammonia borane was removed from the furnace vicinity, and the hBN sample was allowed to cool down within the furnace.

### hBN/NiFeOₓHᵧ anode fabrication

The NiFeOₓHᵧ catalytic layer in our experiments was obtained using the electrochemically deposition (ED) method[52]. First, we prepare Au contacts on quartz substrates (Fig. 1d) using the electron-beam metal deposition technique. Next, the Au contact was immersed in an aqueous solution containing 0.1 M Ni(NO₃)₂·6H₂O and 0.003 M FeCl₂·4H₂O. A potential of −0.55 V (versus Ag/AgCl reference electrode) was applied between the Au contact (as the working electrode for ED) and a carbon counter electrode (Fig. 1d). The deposition of NiFeOₓHᵧ proceeds via the cathodic reduction[52] of NO₃⁻ at the electrode surface which increases the pH to drive metal (oxy)hydroxide precipitation at the electrode surface. The likely deposition mechanism as follows:

$$NO_3^- + 7H_2O + 8e^- \rightarrow NH_4^+ + 10OH^- \qquad (1)$$

$$Ni^{2+} + Fe^{2+} + nOH^- \rightarrow NiFeO_xH_y \qquad (2)$$

The ED was hold at 10 mC cm⁻² until a continuous NiFeOₓHᵧ catalytic layer was obtained. Detailed characterization of the NiFeOₓHᵧ layer can be found in Supplementary Fig. 5.

Subsequently, monolayer hBN was transferred to cover the NiFeOₓHᵧ surface using the wet transfer method[53]. Schematic diagram of hBN-NiFeOₓHᵧ heterostructure assembly can be found in Supplementary Fig. 4. In brief, a thin layer of polymethyl methacrylate (PMMA) was firstly spin-coated on the as-grown hBN/Cu foil and heated at 120 degrees for 20 min. The Cu foil was then etched using a 0.03 g ml⁻¹ (NH₄)₂S₂O₈ solution. After complete etching of Cu substrate, the PMMA/hBN film was washing in deionized water to remove the etchant and ion residues. Cleaned PMMA/hBN was transferred to desired substrate (NiFeOₓHᵧ) and dried for 20 min (120 degree) to enhance the contact between hBN and NiFeOₓHᵧ substrates. PMMA layer was then removed by treating in acetone and isopropanol. Our devices were further encapsulated with electrochemically inert epoxy to ensure that only the hBN-covered areas partake in reactions.

### Material characterizations

Scanning electron microscopy (SEM) and energy dispersive spectrum (EDS) were taken on a GeminiSEM 500 field emission scanning electron microscope operated at 5 kV with EDS detector. SEM, optical microscope (Nikon Eclipse LV150N), and atomic force microscopy (Cypher ES) were used to reveal the surface morphology of the films, and XPS (K-Alpha, Thermo Fisher) was performed to determine their chemical compositions. UV-visible absorption spectrum (MStarter ABS) was measured to estimate the band gap of single-layer hBN transferred to a quartz substrate. High-resolution transmission electron microscope (TEM) and selected area electron diffraction patterns were acquired on JEOL 2100 F at 300 kV from a flat area of the sample suspended on a Gold 300 mesh TEM grid (Quantifoil R1.2/1.3). The HAADF-STEM experiments were performed at 80 kV using FEI Titan Themis G2 300. Gases accumulated in the closed gas circulation system were analyzed by gas chromatography (GC 2010, Shimadzu Co., thermal conductivity detector, Ar carrier gas).

### Electrochemical measurements

All the measurements were carried out on an electrochemical workstation using a custom-made electrochemical cell with a Hg/HgO reference electrode and a Pt counter electrode. The electrolyte was 1 M KOH solution unless otherwise specified. The experimental potentials were converted to the RHE scale using the equation $V_{RHE} = V_{Hg/HgO} + 0.059 \times pH + 0.098$. Linear sweep voltammetry curves

were performed at a scan rate of $1\,mV\,s^{-1}$. All potentials were referenced using the RHE with iR compensation (80% iR drop compensation).

In order to get the effective ECSA of the $hBN/NiFeO_xH_y$ heterostructure, the cyclic voltammetry measurements curves were collected in the potential window between 0.806 and 0.906 V versus RHE at various scan rates from 10 to $100\,mV\,s^{-1}$. By plotting the difference of current density ($J$) between the anodic and cathodic sweeps ($J_{anodic} - J_{cathodic}$)/2 (at 0.856 V versus RHE) against the scan rate, a linear trend was constructed with its slope the double-layer capacitances $C_{dl}$ ($mF\,cm^{-2}$). The ECSA of catalyst is estimated from the $C_{dl}$ according to Eq. 3:

$$ECSA = (C_{dl}/C_s) \times A_{geo} \qquad (3)$$

where $C_s$ (typical 0.04 $mF\,cm^{-2}$) is the specific capacitance, and $A_{geo}$ (1 $cm^2$) is the geometric surface area of the catalyst electrodes[54].

Electrochemical impedance spectroscopy was collected at 1.53 V versus RHE. A sinusoidal voltage with an amplitude of 5 mV and a scanning frequency ranging from 10,000 to 0.01 Hz were applied to carry out the measurements.

### In situ Raman spectra measurements

Raman spectra were collected using a confocal microscope (Horiba LabRAM HR Evolution) with an excitation wavelength of 532 nm. Raman frequency was calibrated by a Si wafer during each experiment. In situ electrochemical Raman experiments were employed in a homemade Raman cell, where a potentiostat was used to control the electrochemical potential.

## Data availability

All data supporting this study and its findings within the article and its Supplementary Information are available from the corresponding author upon reasonable request.

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

## Acknowledgements

The authors acknowledge support from the National Key R&D Program of China (Nos. 2022YFA1505200 and 2018YFA0306900), the National Natural Science Foundation of China (Nos. 21872114, 92163103, and 22171016), and the Fundamental Research Funds for the Central Universities (No. 20720210009).

## Author contributions

Y.L. designed and performed the experiments. B.L. and Y.G. conducted to one-atom-thick hBN synthesis and TEM characterization. N.X. helped with atomic force microscopy tests. Z.Z. conducted the gas permeation measurements. Y.X. helped with e-beam lithography. Y.J. helped with schematic drawing and T.L. helped with the prepared CVD graphene. The manuscript was written by Y.C. with help from S.H. and Y.L. All the authors discussed the results and critically reviewed the manuscript.

## Competing interests

The authors declare no competing interests.
