## [Peer review file · Nature Communications]

REVIEWER COMMENTS

Reviewer #1 (Remarks to the Author):

The authors investigate the utilization of one-atom-thick hBN as a co-catalyst to enhance OER. By encapsulating electrodes with hBN films, they achieve significant improvements in OER efficiency, characterized by ultralow Tafel slopes and high current densities. The hBN co-catalyst exhibits remarkable mass activity, surpassing commercial catalysts by several orders of magnitude. The enhancements observed in this study are attributed to the adsorption of oxygen-containing intermediates and efficient electron transfer at the hBN interface, as revealed through a combination of experimental and simulation studies. These findings shed light on the potential of ultra-thin hBN films as effective co-catalysts for OER, providing valuable insights into atomic-scale interfacial reactions. However, some minor revisions are suggested to improve the clarity and organization of the manuscript.

1. The introduction provides a comprehensive overview of the background and significance of the study. However, it would be beneficial to provide more specific details about the limitations of current electrocatalysts in achieving efficient OER, including any challenges associated with the slow reaction kinetics and the adsorption/desorption of oxygen-containing intermediates. This will help readers better understand the motivation for the research.

2. The paragraph discussing the use of 2D materials as potential catalytic materials lacks clarity and conciseness. It would be beneficial to focus on the specific advantages of atomically-thin 2D materials in shortening electron transport paths and forming heterostructures with other catalytic electrodes via van der Waals interactions. Additionally, it is important to emphasize the need for a deeper understanding of the underlying mechanisms governing OER performance when using 2D materials and their heterostructures.

3. The description of the unique features and characteristics of hBN as a co-catalyst is well-explained. However, it would be helpful to clarify the specific properties that make hBN a potential co-catalyst, such as its insulating nature yet with a thickness down to the mono-atomic limit, which allows for additional charge transport mechanisms. Additionally, providing context for how hBN can modify interfacial adsorption and enhance the OER activity of electrodes would strengthen the explanation.

4. The manuscript presents an innovative co-catalyst for enhancing OER. However, the introduction lacks comprehensive coverage of the crucial aspects such as the role, requirements, and advancements in co-catalyst research. Expanding on these topics would further elucidate the significance of the proposed work and provide better context within the field of OER catalysis.

5. The manuscript provides comprehensive information on the fabrication procedures and structural characterization of the hBN-coated electrodes. However, it is important to address the potential limitations or challenges associated with the synthesis technique, such as variations in film quality or potential impurities, in order to evaluate the reliability and reproducibility of the presented results.

6. The OER performance of the hBN-coated electrodes is impressive, showcasing high current density, long-term stability, and enhanced mass activity. To further strengthen the manuscript, it would be valuable to discuss any potential factors or parameters that may affect the performance, such as the effect of pH, electrolyte composition, or electrode morphology, to provide a more thorough understanding of the observed results.

7. The text mentions that the OER current of approximately 900 mA cm⁻² at VRHE = 1.72 V for the hBN-covered NiFeOxHy electrodes is slightly lower than that of CVD-hBN covered electrodes. Can the authors provide an explanation or speculate on the factors that could contribute to this difference in OER current between the two cases?

8. The text states that a noticeable isotope effect is observed, where the current density obtained in KOH is approximately twice as high as that in KOD. Besides the differences in atomic mass and chemical properties between OH⁻ and OD⁻, are there any other factors that could potentially contribute to this isotope effect? Can the authors provide further analysis to support their conclusion that proton transfer is involved in the rate-determining step of OER reactions?

9. The DFT calculations reveal enhanced adsorption of OH* at B sites with an adsorption energy close

to 1.8 eV compared to that of approximately 0.6 eV on bare NiFeOxHy. Can the authors elaborate on the specific interactions or underlying reasons that lead to this stronger interaction between boron and oxygen atoms and the subsequent weaker binding between O and H atoms in OH* at the hBN/NiFeOxHy electrodes? Additionally, are there any potential implications or consequences of this finding on the overall OER activity and mechanisms at play?

9. The authors emphasize that the electrical resistance of hBN does not restrict the OER activity in their case. However, it would be beneficial for the readers to further understand the role of hBN in enabling fast reactant conversion at high pH. Could the authors provide additional analysis or experimental evidence to support their claim? For example, electrochemical impedance spectroscopy (EIS) measurements or electrical conductivity characterization of hBN films could provide insights into the electron transport mechanism through the atomically-thin hBN layers. Additionally, the authors could discuss the potential contribution of defects or dopants in hBN to the electron transport properties, as these factors may impact the OER activity and overall performance of the hBN/NiFeOxHy electrodes.

10. The in-situ Raman spectroscopy results indicate electron transfer in the NiFeOxHy layer via the valence change of Ni elements. It would be interesting to explore the specific mechanisms through which the change in Ni valency affects the catalytic activity of the hBN/NiFeOxHy electrodes. The authors mention that the change in valency influences the interactions between Ni and adjacent Fe atoms, but further discussion on the nature of these interactions and their impact on the overall OER activity would be valuable. Are there any specific ligand effects or electronic coupling between Ni and Fe atoms that contribute to the observed catalytic enhancement? Furthermore, could the authors discuss any potential differences in the surface structure or composition of the hBN/NiFeOxHy and bare NiFeOxHy electrodes that might account for the distinct mechanisms and resulting activity?

11. The authors introduce one-atom-thick hBN film as an efficient co-catalyst for OER reactions and highlight its little impedance to electron transfer despite being an insulator. Could the authors provide further insights or experimental evidence regarding the mechanisms underlying the low impedance of hBN and its ability to promote electron transfer? Understanding the fundamental principles governing electron transport in hBN would strengthen the scientific foundation of this work. Additionally, it would be valuable to discuss any potential limitations or challenges associated with the integration of hBN as a co-catalyst in practical applications, such as stability under prolonged operation or potential side reactions that may degrade its performance over time.

12. The authors demonstrate that hBN encapsulation is a universal strategy for improving OER performance on various electrodes, including metal (oxy)hydroxides, metals, and carbon-based materials. However, in order to better evaluate the generality and applicability of this strategy, it would be useful to discuss the potential limitations and factors that may influence the magnitude of enhancement observed on different electrode materials. Are there specific conditions or criteria that determine the degree to which OER currents are enhanced by hBN encapsulation? Furthermore, it would be valuable to provide a discussion on the practical implications and prospects of using hBN encapsulation for real-world applications. Are there any scalability, cost, or compatibility challenges that need to be considered when implementing this strategy in large-scale electrochemical devices?

Reviewer #2 (Remarks to the Author):

The manuscript entitled "One-atom-thick hexagonal boron nitride co-catalyst for enhanced oxygen evolution reactions" reported the synthesis of hBN and Ni-Fe (oxy)hydroxide heterostructure catalysts and the application in OER catalysis. The heterostructures did exhibit improved performance. However, in terms of the novelty of research objectives, the approaches and implementation of the characterizations, and the knowledge advances through in-depth mechanical innovations, this work has yet a very far distance to the publication in a high-impact journal such as Nature Communications. Hopefully the following comments could be helpful to further improve the quality of this work.

1. No distinct evidence demonstrates that the obtained materials are monolayer hBN and Ni-Fe (oxy)hydroxides. Multiple and comprehensive characterizations are needed to support the claimed

materials and structures, such as AC-TEM and XRD, otherwise all the calculations, analyses, and discussion based on these materials are incorrect.

2. The monolayer hBN was transferred to cover the NiFeOxHy nanosheets by using a wet transfer method. For this method, the author cited the ref. 40, but it is a review, where no details on the synthesis method were provided, leading to confusions to the readers.

3. What are the bonding states of the heterostructures? No experimental support on this point.

4. The experimental procedures are not very detailed. For example, the electrochemical section lacks a lot of detailed descriptions. In the manuscript, the authors talked the use of different pH values, but there is only 1 M KOH described in the experimental section without any information about the pH values.

5. Calculations are not satisfactory. A set of models should be established for the comparison of adsorption energies, such as separate BN and Ni-Fe (oxy)hydroxides models should be calculated to compare the adsorption energies. The changes in the electronic states after heterostructure coupling and their impact to the catalytic reactions should be discussed in the manuscript.

6. The performance of Ni-Fe (oxy)hydroxides is not as good as other reported literature, please provide a proper explanation.

7. The author used the in-situ Raman to confirm the lower oxidation potential of Ni in the composite. Does the author make the default that the Ni sites are the active sites of OER in the NiFe materials? Ref. 35 cited by the author discussed the catalytic originations of the NiFe alloys, but it is not consistent with the current case.

8. What information could be obtained from the Supplementary Figure 3? Nothing about the NiFeOxHy layer could be read from this figure. Frankly, you can even not say it is NiFeOxHy at all!

9. Based on existing reports, the Ni and Fe after OER reaction should be in a high valence state in the XPS, but the Ni and Fe chemical states in this work remained almost unchanged after tests. Discussion is needed for this point.

10. Some basic electrochemical measurements, such as electrochemical activity active area and EIS, are necessary to better evaluate the electrochemical performance.

11. The author mentioned many advantages of 2D materials in the Introduction, but there are not relevant with the BN and NiFe-(oxy)hydroxide materials studied in this work.

Reviewer #3 (Remarks to the Author):

This manuscript entitled "One-atom-thick hexagonal boron nitride co-catalyst for enhanced oxygen evolution reactions" reported one-atom-thick hexagonal boron nitride (hBN) as an attractive co-catalyst. hBN covered Ni-Fe oxy(hydroxide) exhibited excellent electrochemical OER properties. However, origins of the performance improvement and the mechanism of the OER on hBN covered Ni-Fe oxy(hydroxide) are insufficient for demonstrating the effect of one-atom-thick hBN. Therefore, current form of this manuscript doesn't meet the standard for publishing in this highly impacted journal. Detailed comments are as follows;

- pH-dependent OER kinetics are most likely due to the involvement of lattice oxygen. hBN covered Ni-Fe oxy(hydroxide) is dependent on pH as shown in Figure 2. It appears that lattice oxygen is involved for OER, and if so, the LOM mechanism needs to be elucidated.

- It is necessary to analyze why mechanically exfoliated hBN monolayer crystals have poorer OER performance than hBN via CVD. If it is due to defects, it is necessary to analyze why the defects cause degradation.

- The authors claimed that Fe elements should have little influences to the activity. However, in supplementary 14, Fe effect for improving OER in NiOxHy seems to be large enough.

- The electronic structure of hBN before and after incorporation into NiFeOxHy should be analyzed. Also, the adsorption energy need to be calculated.

REVIEWER COMMENTS

Reviewer #1 (Remarks to the Author):

The authors investigate the utilization of one-atom-thick hBN as a co-catalyst to enhance OER. By encapsulating electrodes with hBN films, they achieve significant improvements in OER efficiency, characterized by ultralow Tafel slopes and high current densities. The hBN co-catalyst exhibits remarkable mass activity, surpassing commercial catalysts by several orders of magnitude. The enhancements observed in this study are attributed to the adsorption of oxygen-containing intermediates and efficient electron transfer at the hBN interface, as revealed through a combination of experimental and simulation studies. These findings shed light on the potential of ultra-thin hBN films as effective co-catalysts for OER, providing valuable insights into atomic-scale interfacial reactions. However, some minor revisions are suggested to improve the clarity and organization of the manuscript.

Response: Thank you very much for your positive comments and valuable suggestions.

1. The introduction provides a comprehensive overview of the background and significance of the study. However, it would be beneficial to provide more specific details about the limitations of current electrocatalysts in achieving efficient OER, including any challenges associated with the slow reaction kinetics and the adsorption/desorption of oxygen-containing intermediates. This will help readers better understand the motivation for the research.

Response: We thank the reviewers for their valuable suggestions and have re-written the introduction accordingly. In the revised manuscript, we have added details about the 4-electron transfer steps involved in OER, as well as the multistep adsorption/desorption of oxygen-containing intermediates (OH*, O* and OOH*) at the electrode-electrolyte interface. Limitations and challenges of current electrocatalysts have also been included.

Please find detailed revision in the first paragraph in the main text.

2. The paragraph discussing the use of 2D materials as potential catalytic materials lacks clarity and conciseness. It would be beneficial to focus on the specific advantages of atomically-thin 2D materials in shortening electron transport paths and forming heterostructures with other catalytic electrodes via van der Waals interactions. Additionally, it is important to emphasize the need for a deeper understanding of the underlying mechanisms governing OER performance when using 2D materials and their heterostructures.

Response: We thank the reviewer for the valuable suggestions. In the revised manuscript, we focus on explaining the specific advantages of 2D materials in shortening electron transport paths. In terms of heterostructures, we focus on discussing the advantages of using 2D heterostructures to construct functional interfaces on-demand [Nature 567, 323–333 (2019)]. The need for mechanism understanding is also further emphasized. By analyzing the current status of 2D materials catalysis, we highlight the importance of creating controlled experimental systems for mechanism studies. We hope these information could provide readers a comprehensive review of the related research fields.

Please find detailed revision in the second paragraph in the main text.

3. The description of the unique features and characteristics of hBN as a co-catalyst is well-explained. However, it would be helpful to clarify the specific properties that make hBN a potential co-catalyst, such as its insulating nature yet with a thickness down to the mono-atomic limit, which allows for additional charge transport mechanisms. Additionally, providing context for how hBN can modify interfacial adsorption and enhance the OER activity of electrodes would strengthen the explanation.

Response: We thank the reviewer for the suggestions. Accordingly, additional information have been added in the revised manuscript, including: 1) the mono-atomic-thick 2D materials allow fast electron tunneling as an additional charge transport mechanism at interfaces; 2) the nature of the B-N bond and its chemical affinity to certain species and 3) previous experimental evidences of using hBN as potential catalysts to emphasize the importance of hBN for tuning interfacial adsorption behaviors.

Please find detailed revision in the third paragraph in the main text.

4. The manuscript presents an innovative co-catalyst for enhancing OER. However, the introduction lacks comprehensive coverage of the crucial aspects such as the role, requirements, and advancements in co-catalyst research. Expanding on these topics would further elucidate the significance of the proposed work and provide better context within the field of OER catalysis.

Response: We thank the reviewer for the helpful suggestions. To that end, in the introduction part, we further review the heterogeneous structure formed between cocatalysts and electrodes, and discuss the optimized mass and energy transfer process at interfaces for enhanced OER performances [*Adv. Mater.* 30, 1704649 (2018), *Angew. Chem. Int. Ed.* 60, 18129-18137 (2021), *J. Am. Chem. Soc.* 140, 12964-12973 (2018)]. Challenges including co-catalysts' contact with substrates, their stabilities and effective activation areas are also discussed.

By adding the above information in the revise manuscript, we feel that now it is more clear of why choosing 2D materials for OER: 1) all atoms there are at surfaces for improved density of active sites; 2) their atomic thickness allows fast electron transfer; 3) 2D materials are impermeable to any species which is essential for improved stability. Thank you again for your helps of improving this work.

5. The manuscript provides comprehensive information on the fabrication procedures and structural characterization of the hBN-coated electrodes. However, it is important to address the potential limitations or challenges associated with the synthesis technique, such as variations in film quality or potential impurities, in order to evaluate the reliability and reproducibility of the presented results.

Response: We appreciate the reviewer's suggestions. In terms of limitations of hBN synthesis, we note that chemical vapor deposition (CVD) method is widely reported for the fabrication of high-quality hBN films. Future optimization direction could be adjusting synthesis parameters including choosing proper precursors, controlling the growth temperatures and time durations, and searching for proper substrates for improved film quality and reduced impurities [*ACS Nano* 7, 5199–5206 (2013), *Nano Lett.* 13, 1834–1839 (2013)]. Other directions include, for example, the

reduction of hBN's cost and developing various synthesis methods such as wet chemistry methods to promote the application of hBN [*Nano Res.* 14, 2424–2431 (2021)]. In terms of heterostructure fabrication, robotic transfer systems will significantly improve the fabrication efficiency. Indeed, in our lab, we have started related research on transferring wafer-scale 2D materials using robotic arms. We would love to share any progress in the near future.

The above discussions have been added in the revised manuscript.

6. The OER performance of the hBN-coated electrodes is impressive, showcasing high current density, long-term stability, and enhanced mass activity. To further strengthen the manuscript, it would be valuable to discuss any potential factors or parameters that may affect the performance, such as the effect of pH, electrolyte composition, or electrode morphology, to provide a more thorough understanding of the observed results.

Response: Thanks for the reviewers' suggestions. Indeed, as you mentioned, a few factors are found to influence the performances, which we detailed below:

1) Solution environment: Fig. 3 shows that increasing the pH of the electrolyte can significantly improve the catalytic activity (Fig. 3a). This result is attributed to the increased amount of reactant (i.e. OH⁻) in solutions. In addition, the OH⁻ also acts as proton accepters, facilitating the deprotonation of OH* which process is the rate determining step in our case. We also note that the type of cations in electrolytes has little effect on the catalytic performance (Supplementary Fig. 10).

2) Electrode morphology: the NiFeO_xH_y electrode we used is amorphous, which is expected to have improved catalytic performance compared with crystalline NiFeO_xH_y due to more active sites [*ACS Catal.* 10, 235–244 (2020)]. The van der Waals contact between NiFeO_xH_y and the hBN layer may also facilitate interlayer charge transfer. [*ACS Catal.* 10, 235–244 (2020), *Nature* 567, 323–333 (2019)]

3) The thickness of hBN crystals: as the thickness of hBN films increases from mono-atomic layer to six layers, the current density of hBN/NiFeO_xH_y catalysts decreases by about 10 times (Supplementary Fig. 17). This can be attributed to the increased electron tunneling barrier in thick hBN films, which scales exponentially with thickness [*Science* 335, 947-950 (2012)].

We have added the above discussion in the revised manuscript at where relevant.

7. The text mentions that the OER current of approximately 900 mA cm⁻² at VRHE = 1.72 V for the hBN-covered NiFeO_xH_y electrodes is slightly lower than that of CVD-hBN covered electrodes. Can the authors provide an explanation or speculate on the factors that could contribute to this difference in OER current between the two cases?

Response: We thank the reviewer for the helpful suggestion that inspires us to carefully investigate the role of exfoliated hBN monolayer crystals. To that end, we have further prepared three hBN/NiFeO_xH_y devices using mechanically exfoliated hBN crystals (Fig. R1). The newly prepared samples show an enhanced OER current of ~2 A cm⁻² at V_{RHE} = 1.72 V, which performance is similar to that from the CVD hBN/NiFeO_xH_y samples. This enhancement indicates that defects in hBN

layers are not likely to be the reactive sites while the basal plane is electro-catalytic.

We also note that the newly prepared samples using exfoliated hBN monolayers have considerably improved performance than that reported in the initial draft because of the following improvements during sample fabrication: 1) it is critical to keep the surface of exfoliated hBN clean. Therefore, the substrates where hBN crystals were mechanically exfoliated were carefully cleaned with isopropanol. 2) the procedures for opening the reaction window on hBN devices were carefully optimized. That includes increased dose for the e-beam lithography, refined subsequent development and cleaning processes to optimize the exposure of the reactive windows.

The above discussion and figures have been added in the revised manuscript and Supplementary Materials.

Figure R1. (a) to (i) Schematic of device fabrication and measurement flow using mechanically exfoliated hBN monolayer crystals. PMMA (polymethyl methacrylate) and PVA (polyvinyl alcohol) substrates are used to improve the cleanliness of the obtained hBN crystal [*Nat. Nanotechnol.* 5, 722–726 (2010)]. (j) Current density diagram of mechanically exfoliated monolayer hBN/NiFeO_xH_y and NiFeO_xH_y samples. Inset shows an optical image of a final device. The area inside the yellow dotted line represents the reaction window defined by e-beam lithography. Error bars represent standard deviations.

8. The text states that a noticeable isotope effect is observed, where the current density obtained in KOH is approximately twice as high as that in KOD. Besides the differences in atomic mass and chemical properties between OH⁻ and OD⁻, are there any other factors that could potentially contribute to this isotope effect? Can the authors provide further analysis to support their conclusion that proton transfer is involved in the rate-determining step of OER reactions?

Response: We thank the reviewer for the comments. Isotopic effects can also be influenced by the difference between proton mobility in water and deuterated water. It has been reported that the proton mobility in deuterated water is 1.6–5.0 times slower than that in water electrolytes [*J. Phys. Chem. Lett.* 8, 3466–3472 (2017)]. Nevertheless, this factor is still incorporated with our conclusion that indeed, proton transfer is the rate-determining step.

We further summarize our logic of concluding that proton transport is the rate-determining step

as follows: First, the pH-dependence (Fig. 3a) indicates that the concentration of either hydroxide ions or protons are critical to the OER activity. Second, we find a pronounced isotope effects between the cases of using KOH (dissolved in H₂O) and KOD (dissolved in D₂O) solutions. The twice higher current density obtained in KOH than that in KOD cannot be attributed to hydroxide ions because of the very similar atomic mass and chemical properties between OH⁻ and OD⁻. The result also indicates that proton transfer is likely to be the dominant factor in OER reactions. Third, density functional theory (DFT) calculations (Fig. 3c) further show that OH* deprotonation is the rate-determining step, which is consistent with the above expectations. Taken all evidences together, we conclude that proton transfer is involved in the rate-determining step of OER reactions.

We have added the above discussion in the revised manuscript.

9. The DFT calculations reveal enhanced adsorption of OH* at B sites with an adsorption energy close to 1.8 eV compared to that of approximately 0.6 eV on bare NiFeO_xH_y. Can the authors elaborate on the specific interactions or underlying reasons that lead to this stronger interaction between boron and oxygen atoms and the subsequent weaker binding between O and H atoms in OH* at the hBN/NiFeO_xH_y electrodes? Additionally, are there any potential implications or consequences of this finding on the overall OER activity and mechanisms at play?

Response: We thank the reviewer for the comments. To further understand the interactions between hBN/NiFeO_xH_y heterostructures and OH species, we calculate the changes in the density of states (DOS) for hBN, NiFeO_xH_y and hBN/NiFeO_xH_y, respectively (Fig. R2). Fig. R2a shows that, after being incorporated with NiFeO_xH_y, the hBN shows a noticeable downward shift of DOS for the anti-bonding orbitals. In addition, an interfacial charge transfer of ~0.15 e⁻ (per 1.33 nm², which is the area used in our model that contains 25 units of BN and 16 units of Ni_{0.75}Fe_{0.25}O_xH_y) from hBN to NiFeO_xH_y is also observed from the charge density calculation of hBN/NiFeO_xH_y heterostructures. These results indicate favorable OH* adsorption on hBN surfaces in hBN/NiFeO_xH_y heterostructures. Therefore, we further calculated the OH* adsorption energy and found that, indeed, the adsorption energy on hBN/NiFeO_xH_y (-1.78 eV) is considerably larger than that on bare hBN surface (-0.92 eV) (Table R1). Such result further supports the importance of hBN/NiFeO_xH_y heterostructures to OER reactions. The strong interaction between boron and oxygen atoms leads to a greater electronic displacement of O atoms toward B atoms [*Angew. Chem. Int. Ed.* e202309158 (2023)]. Therefore, interactions between O and H atoms in OH* are weakened, which behavior considerably reduces the barrier of OH* deprotonation. Since the latter deprotonation is the rate-determining step for the overall OER reactions, improved catalytic performances are observed at electrochemical interfaces that are covered by only one-atom-thick materials.

The above discussions and data have been added in the revised manuscript and Supplementary Materials.

Figure R2. (a) top panel, DOS calculated for individual hBN. Bottom panel, projected DOS of hBN (green line) from the total DOS of hBN/NiFeO_xH_y (red line). (b) top panel, DOS calculated for individual NiFeO_xH_y. Bottom panel, projected DOS of NiFeO_xH_y (green line) from the total DOS of hBN/NiFeO_xH_y (red line).

Table R1. Calculated OH* adsorption energy of hBN, NiFeO_xH_y and hBN/NiFeO_xH_y.

Material	OH* adsorption energy (eV)
hBN	-0.92
NiFeO _x H _y	-0.59
hBN/NiFeO _x H _y	-1.78

10. The authors emphasize that the electrical resistance of hBN does not restrict the OER activity in their case. However, it would be beneficial for the readers to further understand the role of hBN in enabling fast reactant conversion at high pH. Could the authors provide additional analysis or experimental evidence to support their claim? For example, electrochemical impedance spectroscopy (EIS) measurements or electrical conductivity characterization of hBN films could provide insights into the electron transport mechanism through the atomically-thin hBN layers.

Response: We thank the reviewer for the suggestions. Accordingly, the Electrochemical Impedance Spectroscopy (EIS) (Fig. R3) and conductivity characterization of hBN films (Fig. R4) are performed, which results show that indeed the electrical resistance of single layer hBN does not restrict the OER activity in our case.

EIS was measured in 1M KOH (pH=13.65). In Fig. R3, the Nyquist plot is fitted using Randles equivalent circuit model. The charge transfer resistance of hBN/NiFeO_xH_y catalyst (0.4 Ω/cm²) is comparable to that of NiFeO_xH_y (0.9 Ω/cm²) at 1.53 V versus RHE, which indicates negligible interlayer charge transfer impedance due to the presence of hBN. To find out the hBN's conductivity along the direction perpendicular to its basal plane, we use conductive AFM technique where the voltage bias is applied between the AFM tip and the copper substrate (Fig. R4). At small bias (10 mV), a tunneling current of 0.1 nA (note that the contact area of the AFM tip is ~2000 nm²) is found which allows us to estimate an electron conductivity of ~7 x10⁻⁴ S m⁻¹ that is consistent with the values reported in the literature. Further increasing the bias leads to a nonlinear increase of currents which behavior is also a signature of electron tunneling through monolayer hBN crystals

[*Appl. Phys. Lett.* 99, 243114 (2011), *Nano Lett.* 18, 4241–4246 (2018), *Adv. Mater.* 34, 2201387 (2022)]. These results support electron tunneling as the potential electron transport mechanism through hBN.

The above discussion and figures have been added in the revised manuscript and Supplementary Materials.

Figure R3. Comparison of electrochemical impedance spectra (EIS) between NiFeO_xH_y and $\text{hBN}/\text{NiFeO}_x\text{H}_y$. Inset: the electrical equivalent circuit used for EIS fitting, where R_s and R_{ct} denote the electrolyte resistance and the charge transfer resistance, respectively. CPE represents the constant phase elements.

Figure R4. (a) Schematic diagram of the conductive AFM measurements. (b) I - V curves of single layer hBN on Cu substrate. The red dashed line is the best linear fit using the current at small bias (0 to 200 mV).

Additionally, the authors could discuss the potential contribution of defects or dopants in hBN to the electron transport properties, as these factors may impact the OER activity and overall performance of the $\text{hBN}/\text{NiFeO}_x\text{H}_y$ electrodes.

As described in our responses to your 7th questions and Fig. R1, electrodes encapsulated using mechanically exfoliated hBN crystals show a similar OER current density as compared to that from CVD hBN samples, despite the considerably higher defect densities in the latter hBN. This result indicates that defects in hBN layers are not likely to play a noticeable role in our case.

Nevertheless, your idea of using defects and dopants to further tune the OER activity is fundamentally very interesting. We will explore in that direction in the near future.

The above discussions have been added in the revised manuscript.

11. The in-situ Raman spectroscopy results indicate electron transfer in the NiFeO_xH_y layer via the valence change of Ni elements. It would be interesting to explore the specific mechanisms through which the change in Ni valency affects the catalytic activity of the hBN/NiFeO_xH_y electrodes. The authors mention that the change in valency influences the interactions between Ni and adjacent Fe atoms, but further discussion on the nature of these interactions and their impact on the overall OER activity would be valuable. Are there any specific ligand effects or electronic coupling between Ni and Fe atoms that contribute to the observed catalytic enhancement?

Response: We thank the reviewer for the suggestions. To investigate the role of Ni and Fe elements on hBN encapsulated NiFeO_xH_y electrodes, we measured the XPS spectrum of both the hBN/NiFeO_xH_y and bare NiFeO_xH_y electrodes before and immediately after OER reactions. As shown in Fig. R5, the valence state changes of Ni and Fe elements are identical in the two cases, respectively. Therefore, we conclude that Ni and Fe elements still synergistically interact and have similar functions as that in bare NiFeO_xH_y electrodes. The latter functions are widely reported in literatures [*J. Mater. Chem. A*, 10, 23790-23798 (2022), *J. Power Sources* 574, 233163 (2023)]. Such results also suggest the importance of hBN in enhanced OER performance observed in this work.

The above discussion and figures are added in the revised manuscript and Supplementary Materials.

Figure R5. X-ray spectrum of (a) Ni 2p_{2/3} and (b) Fe 2p of NiFeO_xH_y and hBN/NiFeO_xH_y before and after OER reaction. The white and blue areas represent before and after the reaction, respectively.

Furthermore, could the authors discuss any potential differences in the surface structure or composition of the hBN/NiFeO_xH_y and bare NiFeO_xH_y electrodes that might account for the distinct mechanisms and resulting activity?

We believe there are two possible reasons for the distinct mechanisms. First, the lattice oxygen in NiFeO_xH_y is reported to directly participate in OER reactions [*Nat. Commun.* 13, 2191 (2022)] and influence the OH* adsorption status on NiFeO_xH_y. These interactions are screened by the hBN encapsulation in our case, which prevents any direct contact between NiFeO_xH_y and solutions/intermediate species. Therefore, both interfacial mass and charge transfer mechanisms

are different. Second, theoretical calculation results show interlayer charge transfer between hBN and NiFeO_xH_y. The interlayer electrical field promotes the change of Ni's valence state at lower potentials.

The above discussions have been added in the revised manuscript.

12. The authors introduce one-atom-thick hBN film as an efficient co-catalyst for OER reactions and highlight its little impedance to electron transfer despite being an insulator. Could the authors provide further insights or experimental evidence regarding the mechanisms underlying the low impedance of hBN and its ability to promote electron transfer?

Response: We thank the reviewer for the comment. To directly demonstrate the low impedance of hBN, we further performed Electrochemical Impedance Spectroscopy measurements as explained in Fig. R3. The charge transfer resistance of hBN/NiFeO_xH_y catalyst is comparable to that of bare NiFeO_xH_y electrodes, which indicates negligible interlayer charge transfer impedance due to the presence of hBN. We believe the reason for the low resistance is two folds. First, an interfacial electrical field is built between hBN and NiFeO_xH_y layers as explained in Fig. R6, which drives electron transfer at interfaces. Second, the mono-atomic-thickness of hBN allows tunneling to occur as an additional charge transport mechanism [ACS Nano 14, 993–1002 (2020), Appl. Phys. Lett. 99, 243114 (2011)]. To that end, we have measured the electrical conductance of hBN crystals along the direction vertical to the basal plane. As explained in Fig. R4, the result is consistent with the electron tunneling behaviors reported previously.

The above discussions have been added in the revised manuscript and Supplementary Materials.

Figure R6. Calculated charge transfer between hBN and NiFeO_xH_y layers. Cyan and yellow clouds show electron accumulation and depletion near atoms, respectively.

Additionally, it would be valuable to discuss any potential limitations or challenges associated with the integration of hBN as a co-catalyst in practical applications, such as stability under prolonged operation or potential side reactions that may degrade its performance over time.

Response: At laboratory scale, as being demonstrated in our work, the high quality of hBN films ensure the stability of electrodes' performances. At industrial scale, on the other hand, we expect the following potential challenges: 1) meter-sized hBN growth with well-controlled thickness. Otherwise, hBN films that contain areas thicker than monolayer may lead to impeded OER performances. 2) wafer scale transfer methods without introducing cracks and tears on hBN films. Otherwise, the hBN encapsulation is not complete which may cause the degradation of the NiFeO_xH_y layer underneath. The edges at the cracked and teared hBN films are also expected to be highly active due to the unsaturated dangling bonds there [Science 354, 1570 (2016)]. That may

also lead to side reactions and thus stability issues of hBN films.

We have added the above discussion in the revised manuscript.

13. The authors demonstrate that hBN encapsulation is a universal strategy for improving OER performance on various electrodes, including metal (oxy)hydroxides, metals, and carbon-based materials. However, in order to better evaluate the generality and applicability of this strategy, it would be useful to discuss the potential limitations and factors that may influence the magnitude of enhancement observed on different electrode materials. Are there specific conditions or criteria that determine the degree to which OER currents are enhanced by hBN encapsulation?

Response: We thank the reviewer for the suggestions that inspire us to further think about the mechanisms underlying the improved OER performances via hBN encapsulation. As shown in Fig. R7, the enhancement factor (defined as the ratio of current density with and without hBN encapsulation) is found to vary depending on the affinity between OH species and the catalytic electrode materials. For those materials that have relatively weak adsorption energy of OH species, hBN encapsulation significantly improves the OER performance by facilitating the adsorption process. On the other hand, for those materials where OH species adsorption is strong, hBN encapsulation has little influences on the OER performance.

We also note that detailed investigation on this topic requires substantial amount of experimental as well theoretical efforts. Constrained by the length of our manuscript and the focus of hBN/NiFeO_xH_y electrodes in this work, we prefer to leave such detailed investigation as our future works.

Figure R7. Current density diagram of a variety of metal (oxy)hydroxide catalysts. Solid bars are experimental data. The blue and green areas represent strong OH adsorption and weak OH adsorption of metal (oxy)hydroxide, respectively. The blue numbers indicate the enhancement factor for each material. Error bars represent standard deviations. Adsorption energies are obtained from literatures [*Nat. Mater.* 11, 550–557 (2012), *Nat Commun* 11, 2522 (2020), *Angew. Chem. Int. Ed.* 60, 14446-14457 (2021), *Science* 352,333-337 (2016)].

Furthermore, it would be valuable to provide a discussion on the practical implications and prospects of using hBN encapsulation for real-world applications. Are there any scalability, cost, or compatibility challenges that need to be considered when implementing this strategy in large-scale electrochemical devices?

Response: For industrial scale application, we expect the following as potential challenges: 1) meter-sized hBN growth with well-controlled thickness, quality and reproducibility. 2) wafer scale transfer methods that can improve the compatibility between hBN and substrates without introducing cracks and tears on hBN films. Robotic transfer systems for efficient heterostructure fabrication is also required. 3) reducing the cost of hBN films via developing various synthesis methods such as wet chemistry methods to promote the application of hBN.

We have added the above discussion in the revised manuscript.

Reviewer #2 (Remarks to the Author):

The manuscript entitled “ One-atom-thick hexagonal boron nitride co-catalyst for enhanced oxygen evolution reactions ” reported the synthesis of hBN and Ni-Fe (oxy)hydroxide heterostructure catalysts and the application in OER catalysis. The heterostructures did exhibit improved performance. However, in terms of the novelty of research objectives, the approaches and implementation of the characterizations, and the knowledge advances through in-depth mechanical innovations, this work has yet a very far distance to the publication in a high-impact journal such as Nature Communications. Hopefully the following comments could be helpful to further improve the quality of this work.

Response: We thank the reviewer for all the important comments and suggestions that helped us to further improve the quality of our work.

1. No distinct evidence demonstrates that the obtained materials are monolayer hBN and Ni-Fe (oxy)hydroxides. Multiple and comprehensive characterizations are needed to support the claimed materials and structures, such as AC-TEM and XRD, otherwise all the calculations, analyses, and discussion based on these materials are incorrect.

Response: We thank the reviewer for the suggestions. Accordingly, more characterizations about hBN and Ni-Fe (oxy)hydroxides are performed and we hope they can better support our conclusions.

To confirm the structure of the hBN crystal, we further perform the TEM characterization. As shown in Fig. R8, the HAADF-STEM image clearly shows the hexagonal lattice of hBN crystals and distinguishes the B and N atoms where N atoms have a higher intensity than that of B atoms. The STEM-EELS spectrum (Fig. R8c) also shows two peaks around 198.8 and 410.1 eV, corresponding to the K-shell ionization edge of B and N atoms, respectively [*Nat. Electron.* 6, 126–136 (2023), *Materials*, 16, 1864 (2023)]. All these results, together with the XPS (Supplementary Fig. 1a and 1b) and UV–visible spectra (Supplementary Fig. 1c) that we have demonstrated in the initial manuscript, prove the hBN crystal structures.

The next step is to characterize the thickness of the hBN crystal. In addition to AFM measurements (Supplementary Fig. 1d) that we have demonstrated in the initial manuscript, TEM measurements also prove the monolayer nature of our hBN. As shown in Supplementary Fig. 1e and 1f, a fold is found at the edge of the hBN film, with two monolayers clearly seen. Therefore, we believe all these experimental evidences support the monolayer hBN crystal we obtained.

Figure R8. (a) High Angle Angular Dark Field-Scanning Transmission Electron Microscopy (HAADF-STEM) image of a monolayer hBN crystal. The inset shows the corresponding fast Fourier transform. Green and blue balls indicate B and N atoms, respectively. The d -spacings of $(10\bar{1}0)$ planes of hBN is 0.212 nm. (b) Line intensity profile measurements from A-A' in (a) showing the difference in intensities between the B and N atoms. (c) Scanning Transmission Electron Microscopy-Electron energy loss spectroscopy (STEM-EELS) spectrum of hBN.

To find out the structure of Ni-Fe (oxy)hydroxides, we systematically investigate their elements, valence states and structures by the following characterization methods. First, as shown in Fig. R9, X-ray photoelectron spectroscopy (XPS) results confirm the presence of Ni, Fe, and O elements. Their valence states are consistent with those of NiFeO_xH_y catalyst reported in literatures [*Adv. Sci.* 10, 2300717 (2023)]. Specifically, the Ni 2p spectrum shows two peaks that can be assigned to Ni^{2+} , with Ni $2p_{3/2}$ at 855.4 eV and Ni $2p_{1/2}$ at 873.1 eV (Fig. R9a). The Fe $2p_{3/2}$ at 711.5 eV can be assigned to Fe^{3+} [*ACS Catal.* 11, 10537–10552 (2021)] (Fig. R9b). The O 1s characteristic peak at 531.0 and 531.9 eV can be attributed to lattice oxygen and hydroxy groups [*Small Struct.* 3, 2200085 (2022)] (Fig. R9c). In addition, Raman spectrum also proves the existence of Ni-OH, Ni-O and Fe-O bonds in NiFeO_xH_y , which result is consistent with that reported in the literature [*Nat. Commun.* 13, 6094 (2022)]. Specifically, Raman peaks at 450 and 540 cm^{-1} (Fig. R9d) is attributed to vibrations of Ni^{2+} -OH and Ni^{2+} -O, respectively. The broad Raman peak at 680 cm^{-1} can be attributed to Fe-O bonds in the NiFeO_xH_y .

Second, scanning electron microscopy shows that the obtained NiFeO_xH_y layer is uniform with no visible discontinuity. EDS elemental analysis confirms that the Ni, Fe and O elements are uniformly distributed, with a 3:1 atomic ratio between Ni and Fe elements (Supplementary Fig. 5f).

At last, XRD characterization shows no diffraction signals (Fig. R9e), indicating that the NiFeO_xH_y layer is likely to be amorphous. Therefore, we conclude that the material prepared is a layer of amorphous NiFeO_xH_y.

The above discussions and figures have been added in the revised manuscript.

Figure R9. Characterization of the electrochemically deposited NiFeO_xH_y catalytic layer using (a) to (c) X-ray photoelectron spectroscopy of Ni 2p, Fe 2p and O 1s spectrum of NiFeO_xH_y, respectively. (d) Raman spectra and (e) X-ray Diffraction (XRD) techniques.

2. The monolayer hBN was transferred to cover the NiFeO_xH_y nanosheets by using a wet transfer method. For this method, the author cited the ref. 40, but it is a review, where no details on the synthesis method were provided, leading to confusions to the readers.

Response: We thank the reviewer for the helpful suggestion. Accordingly, ref. 40 has been replaced by [Nat. Mater. 21, 740–747 (2022)] where the transfer methods are clearly described. In addition, to avoid any confusion and help readers to reproduce our results, we have added detailed transfer procedures in the revised manuscript with schematics as shown in Fig. R10. In brief, a thin layer of polymethyl methacrylate (PMMA) was firstly spin coated on the as grown hBN/Cu foil and heated at 120 degrees for 20 minutes. The Cu foil was then etched using a 0.03g ml⁻¹ (NH₄)₂S₂O₈ solution. After complete etching of Cu substrate, the PMMA/hBN film was washing in deionized water to remove the etchant and ion residues. Cleaned PMMA/hBN was transferred to desired substrate (NiFeO_xH_y) and dried for 20 min (120 degree) to enhance the contact between hBN and NiFeO_xH_y substrates. PMMA layer was then removed by treating in acetone and isopropanol.

Figure R10. Schematic diagram of wet transfer method.

3. What are the bonding states of the heterostructures? No experimental support on this point.

Response: Thank you very much for the comments. In our heterostructures, the hBN and NiFeO_xH_y layers are physically assembled together via van der Waals interactions. No chemical bonds are expected between them. Experimentally, XPS spectra shows no evidence of additional peaks between the main elements (i.e., B, N, Ni and Fe) after the assembly of hBN on NiFeO_xH_y . This result also supports the absence of bonding between hBN and NiFeO_xH_y layers. From theoretical calculations, the distance between the hBN and NiFeO_xH_y layer is found to be 0.398 nm, in consistent with the typical van der Waals distances of 0.3-0.4 nm [*Nat. Commun.* 12, 91 (2021), *Nature* 567, 323–333 (2019)].

The above discussions and figures have been added in the revised manuscript.

Figure R11. X-ray spectrum (XPS) characterizations of hBN, NiFeO_xH_y and $\text{hBN}/\text{NiFeO}_x\text{H}_y$ heterostructures. (a) Ni 2p and (b) Fe 2p spectrum of NiFeO_xH_y with and without hBN encapsulation. (c) B 1s and (d) N 1s spectrum of hBN before and after assembly on NiFeO_xH_y layers. No peak shift is observed in all cases, indicating the absence of bonding between hBN and NiFeO_xH_y layers

4. The experimental procedures are not very detailed. For example, the electrochemical section lacks a lot of detailed descriptions. In the manuscript, the authors talked the use of different pH values, but there is only 1 M KOH described in the experimental section without any information about the pH values.

Response: We thank the reviewer for pointing out this critical issue. Detailed experimental procedures as explained below have been added in the revise manuscript.

1) For electrochemical measurements, the electrochemical data were collected by employing electrochemical workstation (CHI 750E) at room temperature. All the measurements were carried out on an electrochemical workstation using a custom-made electrochemical cell with a Hg/HgO reference electrode and a Pt counter electrode. The electrolyte was 1 M KOH solution (pH=13.65) unless otherwise specified. The experimental potentials were converted to the RHE scale using the equation:

$$V_{\text{RHE}} = V_{\text{Hg/HgO}} + 0.059 \times \text{pH} + 0.098$$

Linear sweep voltammetry curves were performed at a scan rate of 1 mV s⁻¹. All potentials were referenced using the RHE with *iR* compensation (80% *iR* drop compensation).

2) In pH-dependent measurements, we use KOH solutions with pH values of 11.15, 12.05, 12.90, 13.16, 13.42 and 13.65 at concentrations of 0.001M, 0.01M, 0.1M, 0.2M 0.5M, and 1M, respectively.

The above detailed information has been added in the revised manuscript.

5. Calculations are not satisfactory. A set of models should be established for the comparison of adsorption energies, such as separate BN and Ni-Fe (oxy)hydroxides models should be calculated to compare the adsorption energies. The changes in the electronic states after heterostructure coupling and their impact to the catalytic reactions should be discussed in the manuscript.

Response: We thank the reviewer for the helpful suggestions. Accordingly, we further calculated the OH* adsorption energies of hBN and NiFeO_xH_y, respectively, with results shown in Table R2. hBN/NiFeO_xH_y heterostructures exhibit a higher adsorption energy to OH* than that of bare NiFeO_xH_y or hBN.

Table R2. OH* adsorption energy of hBN, NiFeO_xH_y and hBN/NiFeO_xH_y.

System	OH* adsorption energy (eV)
hBN	-0.92
NiFeO _x H _y	-0.59
hBN/NiFeO _x H _y	-1.78

In addition, we have also calculated the changes in the electronic states in hBN/NiFeO_xH_y heterostructures. A reduced bandgap is found for hBN/NiFeO_xH_y heterostructures with respect to that of either hBN or NiFeO_xH_y (Fig. R12), which result can be attributed to the increased density of states at the Fermi level of NiFeO_xH_y within the heterostructure (Fig. R12b). hBN also exhibits a down-shift in its anti-bonding orbitals (Fig. R12a) which leads to a charge transfer of 0.15 electrons (per 1.33 nm², which is the area used in our model that contains 25 units of BN and 16 units of

Ni_{0.75}Fe_{0.25}O_xH_y) to NiFeO_xH_y. This also explains the high affinity of hBN/NiFeO_xH_y heterostructures to OH species than either of its components (Table R2).

The above discussions and figures have been added in the revised manuscript.

Figure R12. (a) top panel, DOS calculated for individual hBN. Bottom panel, projected DOS of hBN (green line) from the total DOS of hBN/NiFeO_xH_y (red line). (b) top panel, DOS calculated for individual NiFeO_xH_y. Bottom panel, projected DOS of NiFeO_xH_y (green line) from the total DOS of hBN/NiFeO_xH_y (red line).

6. The performance of Ni-Fe (oxy)hydroxides is not as good as other reported literature, please provide a proper explanation.

Response: We thank the reviewer for the suggestion. In literatures, various performances of NiFeO_xH_y catalysts have been reported, depending on the specific morphologies of NiFeO_xH_y as well as that of electrode substrates. For example, the high-performance NiFeO_xH_y catalysts usually have highly porous substrates (Nickel mesh, blank Ni, nickel foam) with a significantly large surface area (around 10 cm²) [*Nat. Commun.* 11, 5075 (2020), *Small Struct.* 3, 2200085 (2022)]. In fact, in our case, if we normalize the current density by the effective electrochemical active surface area (which is estimated to be 1.22 cm²), the activity of our devices per unit area is consistent with that reported in literatures. The mass activity of our amorphous NiFeO_xH_y catalyst reaches 1254 A g⁻¹, which is also comparable to many reported NiFeO_xH_y catalysts (Supplementary Table 1).

The above discussions have been added in the revised manuscript.

7. The author used the in-situ Raman to confirm the lower oxidation potential of Ni in the composite. Does the author make the default that the Ni sites are the active sites of OER in the NiFe materials? Ref. 35 cited by the author discussed the catalytic originations of the NiFe alloys, but it is not consistent with the current case.

Response: We thank the reviewer for the comments. In literatures, the active sites in NiFeO_xH_y is usually attributed to the synergistic interactions between Ni and Fe atoms [*J. Mater. Chem. A*, 10, 23790-23798 (2022), *J. Am. Chem. Soc.* 136, 6744–6753 (2014)]. Our experiments also prove this conclusion. As shown in Fig. R13, adding Fe elements on NiO_xH_y electrodes significantly enhances the catalytic performance.

Figure R13. Current density of NiO_xH_y and NiFeO_xH_y at 1.72 V vs. RHE.

To further investigate the role of Ni and Fe elements on hBN encapsulated NiFeO_xH_y electrodes, we measured the XPS spectrum of both the hBN/NiFeO_xH_y and bare NiFeO_xH_y electrodes before and immediately after OER reactions. As shown in Fig. R14, the valence state changes of Ni and Fe elements are identical in the two cases, respectively. Therefore, we conclude that Ni and Fe elements in hBN/NiFeO_xH_y heterostructures still synergistically interact and have similar functions as that in bare NiFeO_xH_y electrodes. Such results also suggest the importance of hBN in enhanced OER performance observed in this work.

The above discussions and literatures have been added in the revised manuscript.

Figure R14. X-ray spectrum of (a) Ni 2p_{2/3} and (b) Fe 2p of NiFeO_xH_y and hBN/NiFeO_xH_y before and after OER reactions. The white and blue areas represent before and after the reaction, respectively.

8. What information could be obtained from the Supplementary Fig. 3? Nothing about the NiFeO_xH_y layer could be read from this figure. Frankly, you can even not say it is NiFeO_xH_y at all!

Response: We thank the reviewer for the comments and fully accept the criticism. Accordingly, additional characterization including XPS, Raman, and XRD have been performed on the NiFeO_xH_y layer to demonstrate its elements, valence states and bonding structures. Details have been described in our responses to your first question. The updated Supplementary figure is attached below as Fig. R15. We hope it provides information needed for the NiFeO_xH_y layer.

Figure R15. Characterization of the electrochemically deposited NiFeO_xH_y catalytic layer. **(a) to (c)** X-ray photoelectron spectroscopy of Ni 2p, Fe 2p and O 1s spectrum of NiFeO_xH_y, respectively. **(d)** Scanning electron microscope image of the NiFeO_xH_y layer. **(e) to (f)** Energy dispersive spectrum for the elemental distribution in the NiFeO_xH_y layer. **(g)** Optical image of NiFeO_xH_y on Au contact. The NiFeO_xH_y layer thickness is about 60 nm.

9. Based on existing reports, the Ni and Fe after OER reaction should be in a high valence state in the XPS, but the Ni and Fe chemical states in this work remained almost unchanged after tests. Discussion is needed for this point.

Response: We thank the reviewer for the important comments. Accordingly, we have re-performed the XPS measurement on NiFeO_xH_y and hBN/NiFeO_xH_y electrodes before and immediately after the OER reaction to avoid any additional changes of the catalyst materials with time. The updated results are shown in Fig. R16. The valence state changes of Ni and Fe elements are identical in the two cases, respectively. Before OER, the NiFeO_xH_y and hBN/NiFeO_xH_y electrodes show a Ni 2p_{3/2} peak at 855.4 eV and Fe 2p_{3/2} peak at 711.5 eV, which can be attributed to the presence of the Ni²⁺ and Fe³⁺, respectively. While after OER electrolysis, Ni 2p_{3/2} and Fe 2p_{3/2} peaks shifts by 0.9 eV and 0.6 eV toward a higher energy, respectively, suggesting higher oxidation states of the active Ni^{3+δ} and Fe^{3+δ} species [ACS Catal. 11,10537–10552 (2021), J. Phys. Chem. C 120,2247–2253 (2016)].

Thus, the unchanged Ni and Fe chemical states presented in the initial draft is attributed to the long delays of XPS measurements after the OER reactions. According to literatures, the high-valence state metals in NiFeO_xH_y are likely to transit into more stable lower-valence metals [ACS Catal. 10, 235–244 (2020)]. In our case, this transition may further be accelerated by the interfacial

charge transfer from hBN to NiFeO_xH_y. Such transition is also proven by our Raman spectrum measurements, where we observe a Raman peak shift towards lower wavenumbers (Fig. R17) by exposing the hBN/NiFeO_xH_y catalyst in ambient conditions after OER reactions.

The above results and discussions have been updated in the revised manuscript.

Figure R16. X-ray spectrum of (a) Ni 2p_{2/3} and (b) Fe 2p of NiFeO_xH_y and hBN/NiFeO_xH_y before and after electrolysis. The white and blue areas represent before and after the reaction, respectively.

Figure R17. Raman spectra of hBN/NiFeO_xH_y after OER reaction (black line) and leave in the ambient conditions for four hours (red line). The Raman peak of Ni³⁺ (at 466 cm⁻¹ and 550 cm⁻¹) negative shift to (461 cm⁻¹ and 545 cm⁻¹) after 4 hours in the ambient conditions.

10. Some basic electrochemical measurements, such as electrochemical activity active area and EIS, are necessary to better evaluate the electrochemical performance.

Response: We thank the reviewer for the suggestions. Accordingly, we have measured the effective electrochemical active surface area (ECSA) and EIS, with details described below.

In order to get the ECSA of the hBN/NiFeO_xH_y heterostructure, the cyclic voltammetry (CV) measurements curves were collected in the potential window between 0.806 and 0.906 V versus RHE at various scan rates from 10 to 100 mV s⁻¹ (Fig. R18a). A linear dependence is found between

scan rate and the difference of current density between the anodic and cathodic sweeps ($J_{\text{anodic}} - J_{\text{cathodic}}/2$) at 0.856 V vs. RHE. That allows us to obtain the double-layer capacitances C_{dl} (mF cm^{-2}) from the slope (Fig. R18b). The ECSA of catalyst is estimated using the equation:

$$\text{ECSA} = (C_{\text{dl}}/C_s) \times A_{\text{geo}}$$

where C_s (typical 0.04 mF cm^{-2}) is the specific capacitance, and A_{geo} (1 cm^2) is the geometric surface area of the catalyst electrodes [Catalysts 13, 586 (2023)]. The ECSA of the hBN/NiFeO_xH_y is 3.08 cm^2 , and its J_{ECSA} is 60 mA cm^{-2} @1.53V which is an order of magnitude higher than other NiFeO_xH_y catalysts in literature [Nat. Commun. 13, 6094 (2022), Appl. Catal. B. 299,120668 (2021), Small struct. 3, 2200085 (2022)].

Electrochemical impedance spectroscopy (EIS) was collected at 1.53 V versus RHE in 1M KOH. A sinusoidal voltage with an amplitude of 5 mV and a scanning frequency ranging from 10,000 to 0.01 Hz were applied to carry out the measurements. In Fig. R18c, the Nyquist plot is fitted using Randles equivalent circuit model. The charge transfer resistance of hBN/NiFeO_xH_y catalyst is found to be $0.4 \Omega / \text{cm}^2$, which is comparable to that of NiFeO_xH_y found either experimentally in our case ($0.9 \Omega / \text{cm}^2$) or in reported literatures [Adv. Funct. Mater. 32, 2111234 (2022), Adv. Energy Mater. 8, 1703341 (2018)]. This is also consistent with our conclusion that the presence of hBN introduces negligible interlayer charge transfer impedance.

We have added the above discussion and figures in the revised manuscript and Supplementary Materials.

Figure R18. (a) Cyclic Voltammetry (CV) curves for hBN/NiFeO_xH_y carried out in non-faradic regions at different scan rates in 1M KOH. (b) The C_{dl} calculations. (c) Electrochemical impedance spectra (EIS) for hBN/NiFeO_xH_y.

11. The author mentioned many advantages of 2D materials in the Introduction, but there are not relevant with the BN and NiFe-(oxy)hydroxide materials studied in this work.

Response: We thank the reviewer for the comments and have re-written the introduction part accordingly to emphasize the advantages of using hBN and NiFe-(oxy)hydroxide materials for OER reactions.

We summarize the advantages of hBN as follows: 1) its chemical affinity to oxygen-containing species which behavior can optimizes the adsorption of certain species and thus enhances catalytic performances [Nature 494, 455–458 (2013), ACS Catal. 11, 8872–8880 (2021), Science 2016, 354,

1570]. 2) its mono-atomic-scale thickness that allows for fast electron tunneling as an additional charge transport mechanism at interfaces [*Science* 335,947-950 (2012)]. 3) hBN is impermeable to all ions and molecules and thus protects the encapsulated materials for improved stability [*Nat. Commun.* 4, 2541 (2013), *Nat. Nanotechnol.* 11, 218–230 (2016)]. 4) hBN (and in general, 2D materials) can form heterostructures with various 0D, 1D, 2D and 3D catalytic electrodes via van der Waals interactions to extend the possibility of creating functional interfaces on demands [*Nature* 567, 323–333 (2019)]

The reason for choosing Ni-Fe (oxy)hydroxides as our main material is that it is one of the most active OER catalysts [*Small* 16, 2003916 (2020), *J. Am. Chem. Soc.* 137, 1305–1313 (2015), *Adv. Energy Mater.* 8, 1703341 (2018), *Small* 14, 1703257 (2018)]. In fact, devices of hBN encapsulated other materials (including metal (oxy)hydroxide, metal and carbon) have also been investigated as shown in Supplementary.

The above discussions and literatures have been added in the revised manuscript.

Reviewer #3 (Remarks to the Author):

This manuscript entitled “One-atom-thick hexagonal boron nitride co-catalyst for enhanced oxygen evolution reactions” reported one-atom-thick hexagonal boron nitride (hBN) as an attractive co-catalyst. hBN covered Ni-Fe oxy(hydroxide) exhibited excellent electrochemical OER properties. However, origins of the performance improvement and the mechanism of the OER on hBN covered Ni-Fe oxy(hydroxide) are insufficient for demonstrating the effect of one-atom-thick hBN. Therefore, current form of this manuscript doesn't meet the standard for publishing in this highly impacted journal. Detailed comments are as follows;

Response:

We thank the reviewer for all the insightful comments that guide us revise the manuscript.

- pH-dependent OER kinetics are most likely due to the involvement of lattice oxygen. hBN covered Ni-Fe oxy(hydroxide) is dependent on pH as shown in Figure 2. It appears that lattice oxygen is involved for OER, and if so, the LOM mechanism needs to be elucidated.

Response: We thank the reviewer for pointing out the possibility of using LOM mechanism to explain the experimental observations. We would love it if the mechanism is applicable in our case. Indeed, when only the data in Fig. 2 is available, we were also considering the LOM mechanisms. However, after performing hBN permeability experiments, isotope experiments as well as DFT simulations, we have excluded the possibility for the following reasons:

1) Our hBN crystals are proven to be impermeable to ions and gases (Please find evidences in Figure R19 and R20). As a result, the hBN encapsulation prevents any direct contact between the lattice oxygen (in NiFeO_xH_y) to solutions. Therefore, lattice oxygen is not expected to participate in reactions via LOM mechanisms. We would also like to emphasize that the remarkable stability of the hBN/ NiFeO_xH_y system at high current (Fig. 2c in the main text) also indicate the effective encapsulation of hBN and the absence of lattice oxygen involved in reactions. Such involvement

would otherwise lead to compromised catalyst structural integrity and diminished catalytic performance [Joule 5, 1704-1731 (2021)].

Figure R19. Gas permeation measurements proving no Helium transport through hBN membranes. **(a)** Schematics of gas permeation measurements set-up. Black circles represent rubber O-rings for sealing. **(b)** Leak rate (i.e. He permeation rate) as a function of time. We fill the top chamber with the He gas at the time point marked as “Helium on”, and pumped out the He gas at the time point marked as “Helium off”. The permeability of helium through the CVD BN membrane is under our detection limit ($<10^{-14} \text{ mol s}^{-1}$), indicating the membrane porosity $<10^{-6}$.

Figure R20. Ion permeation measurements set-up. The hBN is sandwiched between two reservoirs filled with KOH solutions. No detectable ionic current is obtained within our measurement limit ($\sim 5 \text{ pA}$). That indicates a membrane porosity $<10^{-6}$, which value is consistent with that estimated in gas permeation experiments.

2) To further investigate whether lattice oxygen involved in reactions, we have prepared ^{18}O isotope labelled $\text{NiFe}^{18}\text{O}_x\text{H}_y$ electrodes and encapsulated them using hBN (Fig. R21). OER measures were performed in K^{16}OH dissolved in H_2^{16}O . If LOM mechanisms dominate, ^{18}O element is expected to be found in the gas products. However, this expectation is found against our experiments, where no ^{18}O element higher than nature abundance was detected within our detection limit.

Figure R21. ^{18}O isotope labelling experiment. **(a)** Raman spectrum of ^{18}O -labelled hBN/NiFeO_xH_y electrodes. **(b)** Oxygen products measured by mass spectrum. The relative ratio of $^{36}\text{O}_2$ to $^{32}\text{O}_2$ and $^{34}\text{O}_2$ to $^{32}\text{O}_2$ was measured every 60 s. Dots are experimental data. Solid lines represent their natural abundance.

3) In another control experiments, we have added in solutions tetramethylammonium hydroxide (TMAOH) to poison any lattice oxygen sites if exist [*Nature* 611, 702–708 (2022)]. Negligible reduction of OER currents is observed with respect to that from solutions without TMAOH (Fig. R22), which result further proves that the LOM mechanism is not likely to be the dominant mechanism in our case.

Figure R22. Linear sweep voltammetry of hBN/NiFeO_xH_y in KOH and TMAOH.

The above discussions have been added in the revised manuscript.

- It is necessary to analyze why mechanically exfoliated hBN monolayer crystals have poorer OER performance than hBN via CVD. If it is due to defects, it is necessary to analyze why the defects cause degradation.

Response: We thank the reviewer for the helpful suggestion that guide us to further investigate the performance of exfoliated hBN monolayer crystals. To that end, we have prepared three hBN/NiFeO_xH_y devices using mechanically exfoliated hBN crystals (Fig. R23). The newly prepared samples show an enhanced OER current of $\sim 2 \text{ A cm}^{-2}$ at $V_{\text{RHE}} = 1.72 \text{ V}$, which performance is similar to that from the CVD hBN/NiFeO_xH_y samples. This enhancement indicates that defects in hBN

layers are not likely to be the reactive sites while the basal plane is electro-catalytic.

We also note that the newly prepared samples using exfoliated hBN monolayers have considerably improved performance than that reported in the initial draft because of the following improvements during sample fabrication: 1) it is critical to keep the surface of exfoliated hBN clean. Therefore, the substrates where hBN crystals were mechanically exfoliated were carefully cleaned with isopropanol. 2) the procedures for opening the reaction window on hBN devices were carefully optimized. That includes increased dose for the e-beam lithography, refined subsequent development and cleaning processes to optimize the exposure of the reactive windows.

The above discussion and figures have been added in the revised manuscript and Supplementary Materials.

Figure R23. (a) to (i) Schematic of device fabrication and measurement flow using mechanically exfoliated hBN monolayer crystals. PMMA (polymethyl methacrylate) and PVA (polyvinyl alcohol) substrates are used to improve the cleanliness of the obtained hBN crystal [*Nat. Nanotechnol.* 5, 722–726 (2010)]. **(j)** Current density diagram of mechanically exfoliated monolayer hBN/NiFeO_xH_y and NiFeO_xH_y samples. Inset shows an optical image of a final device. The area inside the yellow dotted line represents the reaction window defined by e-beam lithography. Error bars represent standard deviations.

- The authors claimed that Fe elements should have little influences to the activity. However, in supplementary 14, Fe effect for improving OER in NiO_xH_y seems to be large enough.

Response: We thank the reviewer for pointing out this issue and apologize for any confusion due to our unclear writings. By claiming that “Fe elements should have little influences on the activity”, what we meant is that the hBN cocatalyst mainly promotes the oxidation of Ni²⁺ at lower overpotentials while the Fe elements have similar functions as that in bare NiFeO_xH_y electrodes. To further demonstrate our points, we measured the XPS spectrum of both the hBN/NiFeO_xH_y and bare NiFeO_xH_y electrodes before and immediately after OER reactions. As shown in Fig. R24, the valence state changes of Fe elements are identical in the two cases.

Accordingly, in the revised manuscript we have changed the statement to emphasize the importance of the heterogeneous assembly between hBN and NiFeO_xH_y in enhanced OER performance.

Figure R24. XPS spectra of Fe 2p of NiFeO_xH_y and hBN/NiFeO_xH_y before and after OER reaction. The white and blue areas represent before and after the reaction, respectively.

- The electronic structure of hBN before and after incorporation into NiFeO_xH_y should be analyzed. Also, the adsorption energy need to be calculated

Response: We thank the reviewer for the suggestions. Accordingly, we have calculated the density of states (DOS) for hBN, NiFeO_xH_y and hBN/NiFeO_xH_y, respectively (Fig. R25). Fig. R25a shows that, after being incorporated with NiFeO_xH_y, the hBN shows a noticeable downward shift of DOS for the anti-bonding orbitals. In addition, an interfacial charge transfer of $\sim 0.15 e^-$ (per 1.33 nm², which is the area used in our model that contains 25 units of BN and 16 units of Ni_{0.75}Fe_{0.25}O_xH_y) from hBN to NiFeO_xH_y is also observed from the charge density calculation of hBN/NiFeO_xH_y heterostructures (inset of Fig. 3c and Supplementary Table 3). These results indicate favorable OH* adsorption on hBN surfaces in hBN/NiFeO_xH_y heterostructures [*Science* 307, 403 (2005)]. Therefore, we further calculated the OH* adsorption energy and found that, indeed, the adsorption energy on hBN/NiFeO_xH_y (-1.78 eV) is considerably larger than that on bare hBN surface (-0.92 eV) (Table R3). Such result further supports the importance of hBN/NiFeO_xH_y heterostructures to OER reactions.

In addition, we have also analyzed the changes of DOS in NiFeO_xH_y with and without hBN encapsulation (Fig. R25b). With hBN, an increased DOS is observed at the Fermi level of NiFeO_xH_y which is expected to enhance the conductivity of the NiFeO_xH_y layer and facilitate charge transfer at interfaces.

The above discussions have been added in the revised manuscript.

Figure R25. (a) top panel, DOS calculated for individual hBN. Bottom panel, projected DOS of hBN (green line) from the total DOS of hBN/NiFeO_xH_y (red line). (b) top panel, DOS calculated for individual NiFeO_xH_y. Bottom panel, projected DOS of NiFeO_xH_y (green line) from the total DOS of hBN/NiFeO_xH_y (red line).

Table R3. OH* adsorption energy of hBN, NiFeO_xH_y and hBN/NiFeO_xH_y.

Material	OH* adsorption energy (eV)
hBN	-0.92
NiFeO _x H _y	-0.59
hBN/NiFeO _x H _y	-1.78

REVIEWERS' COMMENTS

Reviewer #1 (Remarks to the Author):

The authors have made comprehensive revisions to the manuscript based on the reviewer's comments and suggestions. I considered that it has met the requirements for publication.

Reviewer #3 (Remarks to the Author):

The authors have satisfactorily addressed the reviewer's comments. The revised manuscript is acceptable for publication in this journal.

Reviewer #4 (Remarks to the Author):

The authors have done an excellent job in addressing all the reviewers comments.

Responses to the reviewers' comments:

Reply to Reviewer #1:

The authors have made comprehensive revisions to the manuscript based on the reviewer's comments and suggestions. I considered that it has met the requirements for publication.

Response:

Thank you very much for your positive comments.

Reply to Reviewer #3:

The authors have satisfactorily addressed the reviewer's comments. The revised manuscript is acceptable for publication in this journal.

Response:

Thank you very much for your positive comments.

Reply to Reviewer #4:

The authors have done an excellent job in addressing all the reviewers comments.

Response:

Thank you very much for your good comments.